# Performance of tumour microenvironment deconvolution methods in breast cancer using single-cell simulated bulk mixtures

Khoa A. Tran [1,2], Venkateswar Addala [1], Rebecca L. Johnston [1], David Lovell[3,4], Andrew Bradley [5], Lambros T. Koufariotis[1], Scott Wood [1], Sunny Z. Wu [6,7], Daniel Roden [6,7], Ghamdan Al-Eryani [6,7], Alexander Swarbrick [6,7], Elizabeth D. Williams [2,8], John V. Pearson[1], Olga Kondrashova [1,9] & Nicola Waddell [1,2,9] ✉

Cells within the tumour microenvironment (TME) can impact tumour development and influence treatment response. Computational approaches have been developed to deconvolve the TME from bulk RNA-seq. Using scRNA-seq profiling from breast tumours we simulate thousands of bulk mixtures, representing tumour purities and cell lineages, to compare the performance of nine TME deconvolution methods (BayesPrism, Scaden, CIBERSORTx, MuSiC, DWLS, hspe, CPM, Bisque, and EPIC). Some methods are more robust in deconvolving mixtures with high tumour purity levels. Most methods tend to mis-predict normal epithelial for cancer epithelial as tumour purity increases, a finding that is validated in two independent datasets. The breast cancer molecular subtype influences this mis-prediction. BayesPrism and DWLS have the lowest combined numbers of false positives and false negatives, and have the best performance when deconvolving granular immune lineages. Our findings highlight the need for more single-cell characterisation of rarer cell types, and suggest that tumour cell compositions should be considered when deconvolving the TME.

The tumour microenvironment (TME) is complex and includes immune cells, blood vessels and stroma[1]. It plays a key role in cancer development, progression and metastasis[2], and the presence of tumour-infiltrating immune cells have been linked to treatment responses and patient outcomes in a variety of cancers[3–5]. For breast cancer, which is a heterogeneous disease comprised of multiple molecular subtypes[6,7], the composition of the TME has varied treatment and outcome implications depending on the molecular subtype.

For example, tumour-infiltrating lymphocytes are predictive of neoadjuvant treatment responses and improved overall survival in triple negative breast cancer (TNBC) and HER2-positive breast cancer, but are associated with an adverse survival in luminal HER2-negative breast cancer[8–11]. Other TME components including tumour-associated macrophages and fibroblasts have also been implicated in influencing breast cancer prognosis and therapy responses[12–14], albeit their contributions are yet to be fully defined.

[1]Cancer Program, QIMR Berghofer Medical Research Institute, Brisbane, QLD 4006, Australia. [2]School of Biomedical Sciences, Queensland University of Technology (QUT), Brisbane, QLD 4000, Australia. [3]School of Computer Science, Queensland University of Technology, Brisbane, QLD 4000, Australia. [4]QUT Centre for Data Science, Brisbane, QLD 4000, Australia. [5]Faculty of Engineering, Queensland University of Technology, Brisbane, QLD 4000, Australia. [6]Cancer Ecosystems Program, Garvan Institute of Medical Research, Darlinghurst, NSW 2010, Australia. [7]School of Clinical Medicine, Faculty of Medicine and Health, UNSW Sydney, Kensington, NSW 2052, Australia. [8]Australian Prostate Cancer Research Centre – Queensland (APCRC-Q) and Queensland Bladder Cancer Initiative (QBCI), Brisbane, QLD 4000, Australia. [9]These authors jointly supervised this work: Olga Kondrashova, Nicola Waddell. ✉e-mail: Nic.waddell@qimrberghofer.edu.au

The introduction of single-cell RNA sequencing (scRNA-seq) has led to a much greater characterisation and understanding of the TME compared to what could be achieved with bulk RNA-seq, across multiple cancers[13,15,16]. Yet, bulk RNA-seq profiling remains a common approach to study the TME, since scRNA-seq is more expensive and requires extensive sample processing not always suitable for analysis of clinical samples or large sample sizes. Several computational methods are available to estimate cell types within the TME by deconvolving bulk RNA-seq data[17–25]. Previous studies that have benchmarked the performance of the TME deconvolution methods have focused either on the technical aspects that could influence deconvolution (e.g. RNA-seq data transformation and normalisation or gene marker selection)[26] or on the overall deconvolution performance[27], and not comprehensively investigating the impact of biological and sample heterogeneity that could influence the TME deconvolution[26–28]. Furthermore, several recently developed methods that utilise scRNA-seq as gene expression reference profiles or as training data are yet to be benchmarked[17,19,24,29].

Different immune and stromal cell subtypes have distinct roles in tumour biology, and can be predictive of treatment responses and outcomes[13,30–34], so accurately discerning these subtypes is critical. Gene expression similarities between cell types of related cell lineages have been shown to affect the performance of transcriptomic-based deconvolution[27]. The high TME deconvolution granularity can in theory be achieved with greater availability of well-annotated scRNA-seq datasets and the recently developed scRNA-seq-based deconvolution methods[23,29], yet the performance of such granular deconvolution still needs to be benchmarked. Additionally, factors such as tumour purity (i.e. proportion of tumour cells), which can vary greatly between samples, may influence TME deconvolution performance[18].

In this study, we comprehensively benchmark the impact of variable tumour purity, absent cell types and lineages of epithelial and immune cell types on the performance of computational TME deconvolution. We evaluate the performance of three distinct groups of recently developed deconvolution methods: seven single-cell-based methods (CIBERSORTx[19], MuSiC[20], Bisque[22], DWLS[23], CPM[24], BayesPrism[29], hspe[35]), one signature-genes-based method (EPIC[25]), and one deep-learning-based method (Scaden[17]). We use a scRNA-seq breast cancer atlas to simulate artificial bulk RNA mixtures with known purity levels and granular compositions of tumour and normal epithelial cells, B-cells, T-cells and myeloid subtypes to determine the impact of sample heterogeneity on the performance of each deconvolution method.

## Results

### Simulation of artificial bulk mixtures to assess performance of TME deconvolution methods

To test the performance of nine TME deconvolution methods, we simulated artificial bulk RNA-seq mixtures using published scRNA-seq breast cancer data[15]. The scRNA-seq data was derived from 26 breast cancer patients representing three molecular subtypes: ER+ ($n = 11$ patients), HER2+ ($n = 5$ patients) and triple-negative breast cancer (TNBC; $n = 10$ patients). In total, 100,064 cells were annotated into nine major cell types, 29 minor cell types and 49 subsets (Fig. 1a and Supplementary Data 1). Each patient sample had differences in cell abundance per cell type, with some cell types absent from specific patients and present cell types ranging from one cell (most rare cell types) to 4596 cells (most abundant cell type; Supplementary Data 1). To ensure an even representation of cell types within each patient and enable a diverse range of simulated mixtures, an oversampling method called Synthetic Minority Oversampling Technique (SMOTE)[36], was used to synthesise between 25 and 4575 cells per cell type for each patient sample. Of note, we discarded cell types with less than 10 cells to ensure SMOTE has a reasonable pool of original cells. The synthesised

data resulted in the number of cells in each cell type that was present in a sample, matching the number of cells in the most abundant cell type for that sample (Supplementary Fig. 1 and Supplementary Data 2). The gene expression profiles of synthesised cells were consistent with those of the original cells (Supplementary Fig. 2).

To generate the bulk mixtures from the single cell data, we split the data into training (18 patient samples) and test (8 patient samples) data sets, ensuring that all three breast cancer subtypes (ER+, HER2+ and TNBC) and major cell types were represented in both sets (Fig. 1b, Supplementary Data 1). We used a sparse simulation process to create the bulk cell mixtures. This approach randomised the number of cell types and enabled a much more diverse proportion range across all cell types compared to previously used resampling[17] (Supplementary Fig. 3 and Methods), to simulate bulk RNA-seq data with varying tumour purity and immune cell lineages to assess performance of the nine deconvolution methods (Fig. 1b, c).

### Performance of TME deconvolution methods across tumour purity levels

To assess the impact of variable tumour purity levels (proportion of tumour cells) on the performance of the TME deconvolution methods, we simulated 38,000 test cell mixtures comprised of 2000 simulations (250 per patient) for each of the 19 purity levels, ranging from 5% to 95% tumour cells (Fig. 1b). Bray-Curtis dissimilarity between predicted and true proportions across all cell types suggested that BayesPrism, Scaden, and MuSiC outperformed other methods across all purity levels, evidenced by the lowest dissimilarity scores across (Fig. 2a and Supplementary Fig. 4a). Among them, Scaden achieved the lowest Bray-Curtis dissimilarity scores for tumour purity levels below 15% (≤ 0.13) and at 95% (0.08), and BayesPrism achieved the lowest dissimilarity scores for all other tumour purity levels. Moreover, BayesPrism, MuSiC, and hspe generally performed better in samples with higher tumour content, evidenced by decreasing Bray-Curtis dissimilarity, while DWLS, CBX, Bisque, EPIC, and CPM performed worse with higher tumour purity levels, evidenced by increasing Bray-Curtis dissimilarity. BayesPrism, Scaden and MuSiC also had the most stable and highest median Pearson's $r$ values (≥0.86 for BayesPrism, ≥0.87 for Scaden, and ≥0.79 for MuSiC) across all tumour purity levels (Supplementary Fig. 4b). The poorest overall performance was observed for Bisque, EPIC and CPM. CPM was the worst performing method, with the highest Bray-Curtis dissimilarity (0.43–0.83) and lowest Pearson's $r$ (≤0.3) for all tumour purity levels.

To investigate whether prediction of specific cell types was impacted by tumour purity, we analysed the median RMSE values for the nine major cell types. In terms of the performance of each method to predict the three major immune cell types (T-cells, B-cells and myeloid cells), BayesPrism, Scaden, MuSiC, CBX, and DWLS were the only methods that achieved RMSE values < 10 for all three immune cell types for mixtures across all tumour purity levels (Fig. 2b and Supplementary Fig. 4c). BayesPrism and DWLS were superior to Scaden, MuSiC, and CBX in deconvolving T-cells and B-cells, evidenced by lower RMSE values at all purity levels. Notably, DWLS achieved the lowest RMSE values for B-cells across all purity levels. In contrast, Bisque, hspe, EPIC and CPM obtained RMSE values ≥ 2 for the immune cells at most tumour purity levels and ≥10 at low tumour purity levels, suggesting relatively poor performance for immune cells (Fig. 2b and Supplementary Fig. 4c). Of note, CPM predicted similar proportions for each cell type regardless of tumour purity level (Supplementary Fig. 5).

Cancer and normal epithelial were the most mis-predicted cell types (highest RMSE values) across all purity levels for BayesPrism, Scaden, MuSiC, CBX, DWLS, hspe, and EPIC (Fig. 2b and Supplementary Fig. 4c), with the magnitude of mis-prediction increasing with

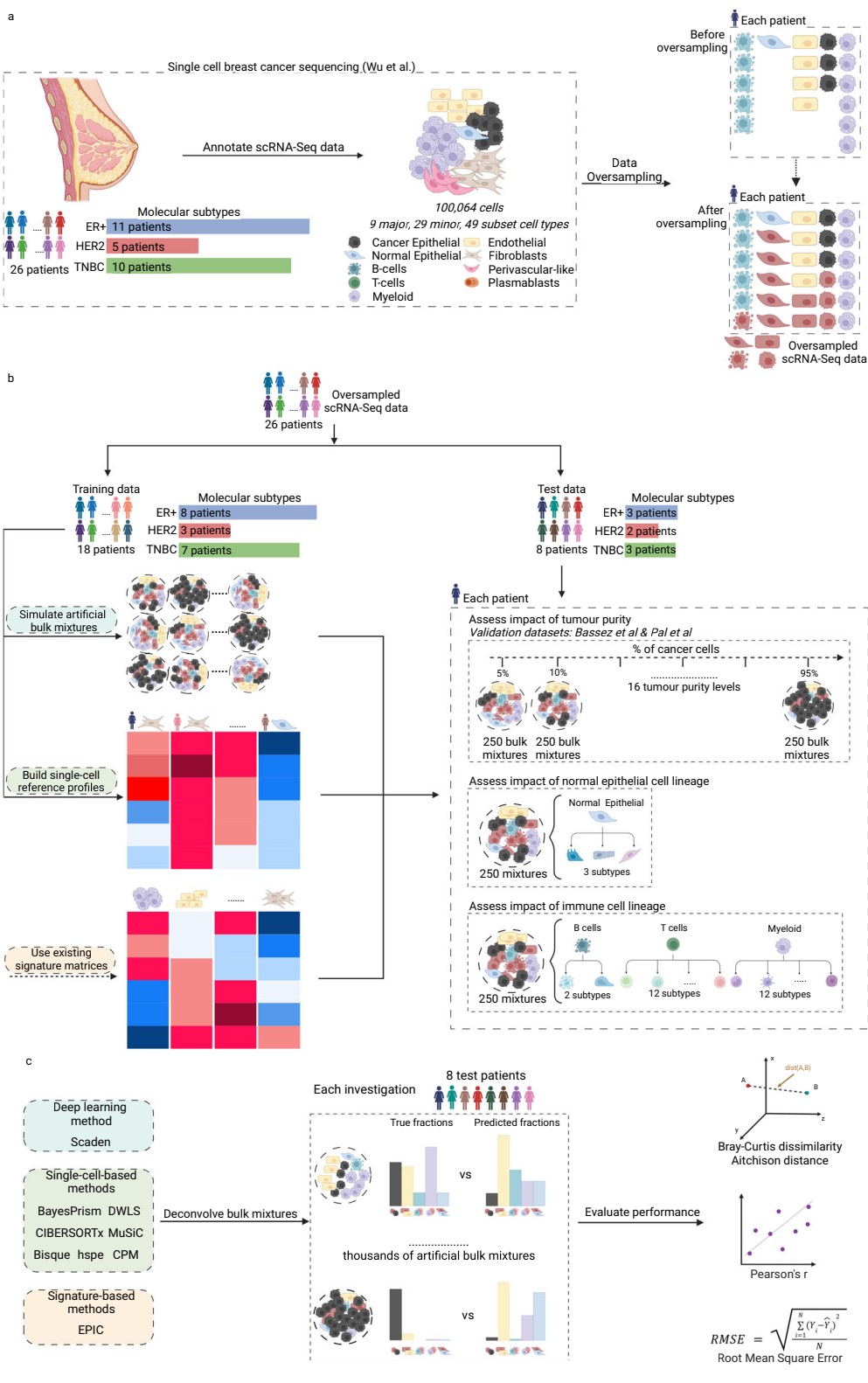

Created with BioRender.com

higher tumour purity (indicated by higher RMSE values). Interestingly, raw prediction errors, which capture directionality in error distributions of cancer epithelial and normal epithelial cell types, showed that cancer and normal epithelial cells were mis-predicted at similar magnitudes, suggesting that the cancer epithelial cell type was deconvolved incorrectly as normal epithelial in cell mixtures with higher tumour content (Supplementary Fig. 6).

## Confirmation of the performance of TME deconvolution methods

We used two additional single-cell RNA-seq datasets, Bassez et al.[37] and Pal et al.[38], to repeat the analysis that assessed the impact of tumour purity on deconvolution. Using this data, we simulated a further 85,000 test cell mixtures comprised of 5000 simulations (250 per patient) for each of the 19 purity levels.

**Fig. 1 | Experimental design of the benchmarking study.** Workflow to benchmark performance of nine transcriptomic-based TME deconvolution methods in different biological conditions using scRNA-seq breast cancer data. **a** Annotated scRNA-seq data from Wu et al.[15] is oversampled so that within each patient, the number of cells in the less abundant cell types matches the number in the most abundant cell type. **b** Oversampled scRNA-seq data are assigned to train data ($n = 18$ patients) and test data ($n = 8$ patients). Train data was used to generate either artificial bulk mixtures or single-cell reference matrix as input to the different TME deconvolution methods (left block). Test data was used to generate artificial bulk mixtures for different benchmarking investigations (tumour purity, normal epithelial cell lines, immune cell linage) (right block). **c** Within each investigation, the overall deconvolution performance of the nine benchmarked methods was evaluated using Bray-Curtis dissimilarity, Aitchison distance, RMSE and Pearson's r, while the performance of predicting individual components was assessed using RMSE. ER+: estrogen receptor positive, HER2+: Human Epidermal growth factor Receptor 2 positive, TME: tumour microenvironment, TNBC: triple-negative breast cancer, RMSE: Root Mean Square Error, scRNA-seq: single-cell RNA sequencing. Figure was made using BioRender.com.

In agreement with the initial findings from the simulated data from Wu et al.[15], BayesPrism showed the best overall performance and robustness against tumour purity, being the only method achieving Bray-Curtis dissimilarity ≤0.22 across all tumour purity levels for simulated mixtures from both datasets (Supplementary Fig. 7a, b for Bassez et al.[37], and Supplementary Fig. 7c, d for Pal et al.[38]). Scaden, MuSiC and CBX also showed good performance with Bray-Curtis dissimilarity scores ≤0.22 for Scaden, ≤0.25 for MuSiC, and ≤0.37 for CBX across both datasets. Among the other 4 methods (Bisque, EPIC, DWLS and hspe), only DWLS achieved comparable performance to BayesPrism, MuSiC and Scaden (Bray-Curtis dissimilarity ≤0.16), but only with mixtures from Pal et al.[38].

BayesPrism, Scaden, MuSiC, CBX, and DWLS exhibited good performance on immune cell types, consistent with findings from the Wu et al.[15] dataset, (Supplementary Fig. 8a for Bassez et al.[37], and Supplementary Fig. 8b for Pal et al.[38]). In terms of cancer-normal misprediction, the Bassez et al.[37] dataset did not have normal epithelial annotations, however the data from Pal et al.[38] confirmed that MuSiC, CBX, and DWLS showed misprediction of cancer and normal epithelial that increased with tumour purity (Supplementary Fig. 8b). Interestingly, while Bisque performed better with cancer populations at lower tumour levels in Wu et al.[15] (Fig. 2b and Supplementary Fig. 4c) and Bassez et al.[37] datasets (Supplementary Fig. 8a), this phenomenon was reversed in the Pal et al.[38] dataset (Supplementary Fig. 8b). We investigated this further by examining the distributions of predicted cancer proportions, aggregated by tumour purity levels (Supplementary Fig. 9). The results showed that most of Bisque predictions centred around 20% tumour purity in Wu et al.[15] and Bassez et al.[37] datasets, and dramatically shifted 65% in the Pal et al.[38] dataset. By contrast, predicted cancer proportions of the top performing methods (BayesPrism, Scaden, and MuSiC) generally followed their corresponding tumour purity levels, highlighting the robustness of these methods across datasets.

Additionally, we have tested the robustness of deconvolution performance when technical batch effect is present by using single-cell reference profiles from Wu et al.[15] for deconvolution of simulated mixtures generated from Bassez et al.[37] (Supplementary Fig. 10a, b) and Pal et al.[38] (Supplementary Fig. 10c, d). To ensure cell-type labels are consistent across all three datasets, we collapsed several cell types for Bassez et al.[37] and Pal et al.[38], e.g. dendritic cell, macrophage and myeloid are grouped into myeloid (see Methods/Datasets and preprocessing). Except for a reduction in performance of hspe in Pal et al.[38], results were consistent with patterns observed in the Wu et al. dataset mixtures (Fig. 2a, b, Supplementary Fig. 4).

To confirm the deconvolution performance for major cancer and immune cell types using non-simulated real bulk RNA-seq data, we used data derived from breast cancer patients from The Cancer Genome Atlas (TCGA) study[6,39]. We compared predicted cancer cell proportions against Consensus Purity Estimates (CPE) produced in Aran et al.[40] ($n = 1031$), and predicted lymphocyte proportions (T-cells and B-cells) against tumour-infiltrating lymphocytes (TIL) estimates produced from H&E images using deep learning[41] ($n = 892$). Consistent with the simulated bulk results, BayesPrism, Scaden and MuSiC showed the strongest performance for both cancer and lymphocyte predictions, evidenced by the highest Pearson's correlation

coefficients and lowest RMSE scores (Fig. 2c, d). Similar to simulated bulk results, CBX, DWLS and hspe achieved better performance than Bisque, EPIC and CPM in predicting cancer proportions (Fig. 2c), however, except for hspe and DWLS, the 4 methods over-predicted lymphocyte proportions (Fig. 2d). Generally, we did not observe patterns in prediction correlations across the PAM50 subtypes.

Additionally, to confirm that cancer epithelial cells are mispredicted as normal epithelial in the simulated bulk experiments, we collapsed cancer epithelial and normal epithelial cell types into one class termed Epithelial and predicted major cell types. Compared to previous predictions when all major cell types were considered (Supplementary Fig. 4c), RMSE values decreased to ≤10.03, 10.75, 6.13, 15.53, 11.5, 10.31 and 15.45 for BayesPrism, Scaden, MuSiC, CBX, DWLS, hspe and EPIC, respectively (Supplementary Fig. 11a). Similarly, when we simulated additional 2000 mixtures (250 per patient) per purity level with no normal epithelial cells, we observed decreased RMSE values for cancer epithelial cell type predictions for the same seven methods (Supplementary Fig. 11b).

## Performance of deconvolution methods across normal epithelial lineages in different breast cancer molecular subtypes

Next, we explored whether the mis-prediction of cancer epithelial cells as normal epithelial cells (observed in the seven methods) was associated with breast cancer molecular subtypes (ER+, HER+ and TNBC) or normal epithelial minor cell types (luminal progenitors, mature luminal and myoepithelial). To study this two-factor relationship, we simulated new mixtures using minor subtype annotations for normal epithelial cells at a fixed tumour purity of 50%, which was selected as most methods perform optimally at this purity level (Fig. 2b). Within each breast cancer molecular subtype, cancer epithelial and normal epithelial minor cell types were the main drivers of raw predictions errors, and the prediction of the three normal epithelial minor cell types varied (Fig. 3a, b). In TNBC, luminal progenitors produced highest RMSE scores (Fig. 3a) and were over-predicted while cancer epithelial was under-predicted (Fig. 3b) across all deconvolution methods, suggesting they are the likely cause of normal epithelial cell mis-prediction. In contrast, the mature luminal cells were the likely cause of mis-prediction in ER+ tumours. For the HER2+ molecular subtype, the prediction errors were elevated for either luminal progenitors or mature luminal cells depending on the method (luminal progenitors for BayesPrism, Scaden, MuSiC and DWLS; mature luminal for EPIC; and both for CBX).

## False positive and false negative predictions across deconvolution methods

In the context of deconvolution, false positive and false negative predictions can lead to severe mischaracterisation of cell compositions within the TME. False positives occur when a method predicts a cell type to be present, while it is absent in the mixture (<0.1%), and zero false negatives occur when a method predicts a cell type to be absent (< 0.1%), while it is present (Fig. 4a). We used the 2000 cell mixtures (250 per patient) at 50% tumour purity and nine major cell types from the previous tumour purity experiment (Fig. 1b) to determine false positive and false negative prediction rates for each deconvolution method.

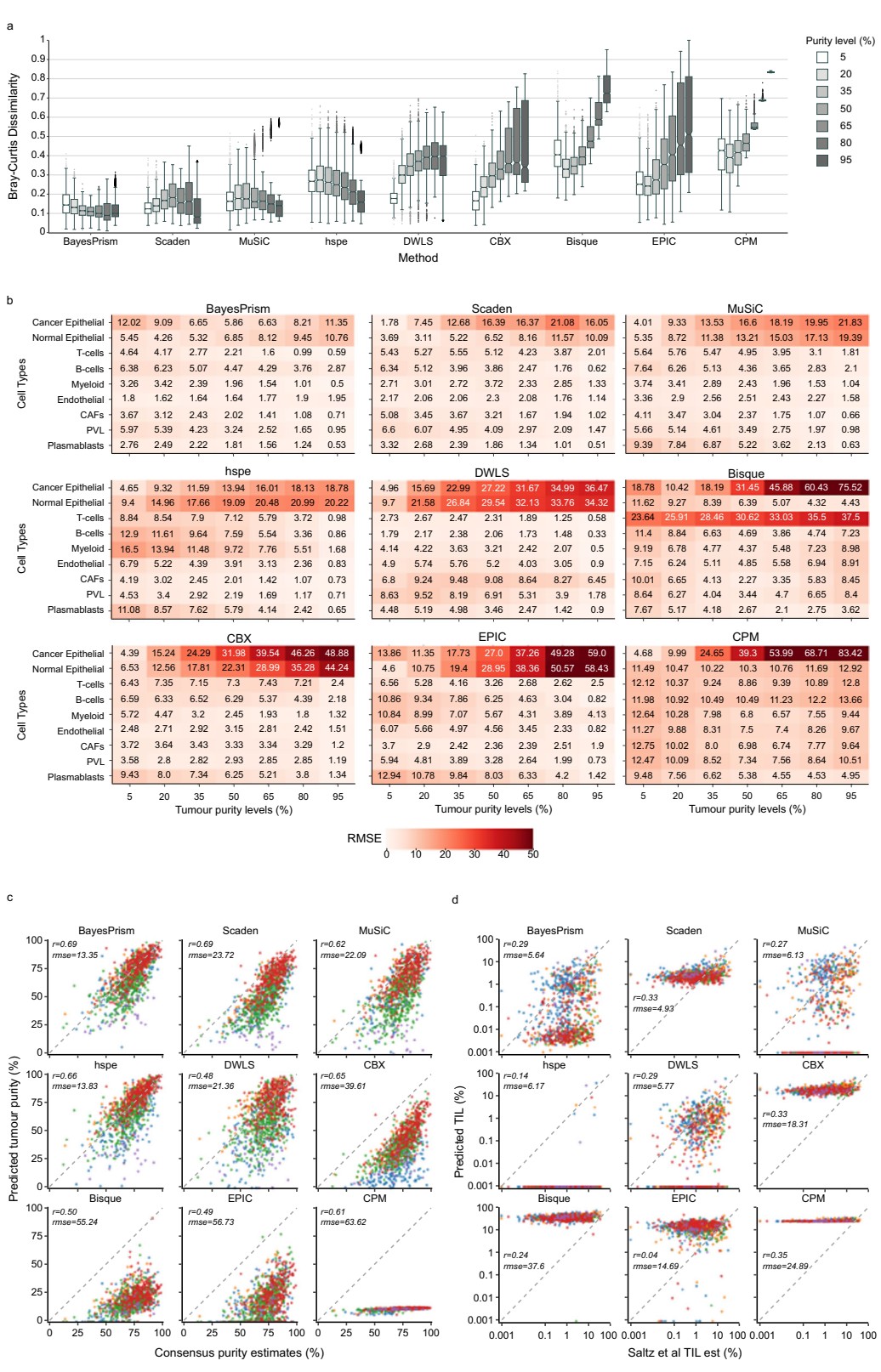

For the false positive predictions, we focused on cell mixtures where one or more cell-type components were absent (*n* = 5247 of 18,000 components across 2000 mixtures each with nine components). Overall, the hspe method predicted the lowest percentage of false positives across all cell types (20.7%), followed by BayesPrism (31.9%), MuSiC (36.1%), EPIC (46%), DWLS (48.4%), CBX (50%) and Bisque (61.1%) (Fig. 4b). Scaden and CPM showed the highest false

positive rate. For Scaden, most false positives (74.1%) were mispredicted to be contributing between 1–10% to a mixture, for CPM the majority of false positives (69%) were predicted to be contributing over 10% (Supplementary Fig. 12a). The CPM result was expected, since as noted earlier CPM predicted similar proportions for each cell type regardless of mixture composition (Supplementary Fig. 5). The cell type with the largest proportions of false positives was normal

**Fig. 2 | Impact of variable tumour purity on deconvolution. a** Bray-Curtis dissimilarity between predicted and ground truth cell compositions across 7 tumours purity levels (from 5% to 95%, 15% interval). Deconvolution methods are organised in order of decreasing performance based on their median Bray-Curtis dissimilarity values. $n = 2000$ artificial bulk at each purity level. Each box represents the middle 50% of Bray-Curtis values, which includes the first quartile (Q1), the median, and the third quartile (Q3). Upper and lower whiskers depict maxima and minima of Bray-Curtis values, excluding outliers. Outliers are Bray-Curtis values that are more than 1.5x the interquartile range from either Q1 or Q3. Higher Bray-Curtis dissimilarity indicates poorer performance. **b** Median RMSE between predicted and actual cell compositions, aggregated by cell type. Seven tumour purity levels are shown (from 5% to 95%, 15% interval). Darker shade of red represents higher RMSE values and poorer performance, with numeric RMSE values shown. Major cell types (y-axis) are organised into three categories: epithelial (normal epithelial and cancer epithelial), immune (T-cells, B-cells and myeloid), and stromal cells (endothelial, CAFs, PVL and plasmablasts). CAFs: Cancer Associated Fibroblasts, PVL: Perivascular-like, RMSE: Root Mean Square Error. Scatter plots of predicted tumour purity (cancer epithelial proportions, y-axis) versus tumour purity derived from copy number variations by Aran et al.[40] (x-axis) in linear scale **(c)**, and predicted lymphocytes (T-cells and B-cells, y-axis) versus tumour-infiltrating lymphocytes (TIL) estimations by Saltz et al.[41] (x-axis) in log scale **(d)**. Each point represents one bulk mixture from TCGA breast cancer patient, with its colour representing the associated molecular subtypes. Dotted 45-degree diagonal line represents perfect prediction where predicted proportions match actual proportions. Each subplot is annotated with its correlation coefficient (*r*) and root mean square error (*rmse*). Source data are provided as a Source Data file.

epithelial with most severe mis-predictions across all methods except for Bisque (Fig. 4). In terms of the proportion of false positives for the immune cell types (T-cells, B-cells and myeloid cells), hspe, BayesPrism, MuSiC, and DWLS showed the best performance with false positive rates of ≤42% (Fig. 4b) and the majority of mis-predicted proportions being either 0.1–1% or 1–10% (Supplementary Fig. 12a). The false positive rate was not determined for cancer epithelial cell type, as the tumour purity was fixed at 50% for this experiment.

For the false negative rate calculation, we focused only on the cell-type components that were present in the mixtures ($n = 12,753$ of 18,000 components). In terms of false negatives, Scaden and CPM were the only methods with no false negatives, but in the case of CPM (Fig. 4c), this was caused by poor overall performance with similar proportions predicted for each cell type. Among the methods with detected false negatives, BayesPrism, MuSiC, CBX, and DWLS had the lowest overall false negative rate of 2.6%, 5.7%, 3.2%, and 6.8% respectively (Fig. 4c). While hspe showed the best performance for false positives, it had the highest false negative rate of 24.4%. When compared by cell type, Bisque was the only method with a high false negative rate for the normal epithelial cell type (40.3%; Supplementary Fig. 12b). In terms of false negatives for the immune cell types, BayesPrism, MuSiC, CBX, and DWLS showed the best performance. Notably, Bisque did not predict any false negatives for T-cells but had a high false negative rate (11.5%) for B-cells, most with the ground truth proportions between 1–10% (Supplementary Fig. 12b). Overall, taking both false positive and negative rates into consideration, no single method outperformed the others, but BayesPrism, MuSiC, CBX and DWLS showed the best comparable performances.

**Performance of deconvolution methods across lineage levels of immune cells**

Our initial observations suggested that two methods, BayesPrism and DWLS had the best performance in deconvoluting major immune cell types, followed by Scaden, MuSiC and CBX (Fig. 2b). We next sought to determine whether the deconvolution performance would decline in the context of minor cell types or more granular subset cell types for T-cells (11 subtypes), B-cells (2 subtypes), and myeloid cells (10 subtypes; Fig. 5a).

We used Aitchison Distance to compare the overall performance of each method across lineage levels. BayesPrism had the lowest median distance corresponding to the best overall performance at all major, minor and subset cell type levels, with Aitchison distance of 2.88, 8.2 and 12.14 at major, minor and subset level, respectively (Fig. 5b). The overall performance across three levels was followed by DWLS, MuSiC, and CBX. While Scaden performance was comparable to the top four methods at major and minor level (3.86 and 7.53, respectively), its' performance severely deteriorated at subset level (15.46), dropping to the second worst performance after CPM (24.41; Fig. 5b). When using only the immune portion of the mixtures to calculate the Aitchison distance between predicted and expected proportions, DWLS outperformed the other methods at subset and minor

levels, while BayesPrism remained the best method at major level (Fig. 5c). For the subsequent analysis, we focused on BayesPrism and DWLS, as these methods showed the best and second-best overall performance across all lineage levels, respectively.

For both, BayesPrism and DWLS, scatter plots of predicted versus actual proportions showed higher level dispersion, indicating worse performance, with increased level of granularity for T-cells, B-cells and myeloid (Supplementary Fig. 13). This was supported by analysis of the raw prediction errors, which showed that major cell type predictions had the least number of outliers, and minor and subset cell type predictions had more outliers compared to their parent major and minor cell types, respectively (Supplementary Fig. 14). With regards to deconvolving immune cell types at each lineage level, the performance of BayesPrism was worse than DWLS across major (RMSE 2.0-4.5 for BayesPrism and 2.1-3.2 for DWLS), minor (RMSE 3.4-8.5 for BayesPrism and 2.8-9.4 for DWLS) and subset levels (RMSE 0.8-9.4 for BayesPrism and 0.7-14.8 for DWLS; Fig. 5d, Supplementary Data 3). For both BayesPrism and DWLS, several cell types at minor and subset levels (e.g. NKT cells, B-cell naïve cells at minor level, and IFN-Signature T-cells and Mono:IL1B cells at subset level) showed patient-specific clustering of predictions (Supplementary Fig. 13).

At minor and subset levels, small errors can be more severe than at major level as cell-type proportions are generally low. We used the Relative Proportion Error (RPE) values to understand the magnitude of misprediction per each percentage of ground truth. Under this criterion, DWLS produced lower RPE values than BayesPrism for most cell types at all three lineages levels (Fig. 5e, Supplementary Data 3). However, both BayesPrism and DWLS also produced several extreme mispredictions at subset levels, for example RPE values over 24 for cycling T-cells and chemokine-expressing T-cells for BayesPrism, and RPE values over 25 for cycling T-cells and Mono:FCGP3A for DWLS.

BayesPrism and DWLS produced false positive and false negative predictions at the minor and subset lineage levels (Supplementary Fig. 15a, b). Overall, DWLS and BayesPrism had the lowest false positive rates across all minor and subset immune cell types, respectively (29.9% for DWLS and 22.10% for BayesPrism; Supplementary Fig. 15a). For both methods, most false positives were mis-predicted to be contributing between 1–10% and above 10% for all minor and subtype cell types. On the other hand, DWLS outperformed BayesPrism in handling false negatives, achieving lower false negatives rates at both minor (28.3% compared to 40.1% of BayesPrism) and subset (49.0% Compared to 55.8% of BayesPrism) levels (Supplementary Fig. 15b). Similar to false positives, most false negatives were mis-predicted to be contributing between 1–10% and above 10% for all minor and subset cell types.

## Discussion

Accurately profiling the TME provides insight into tumour development and can lead to identification of prognostic and treatment markers. Single-cell sequencing and spatial transcriptomics enable high-resolution profiling of the TME, however these technologies are

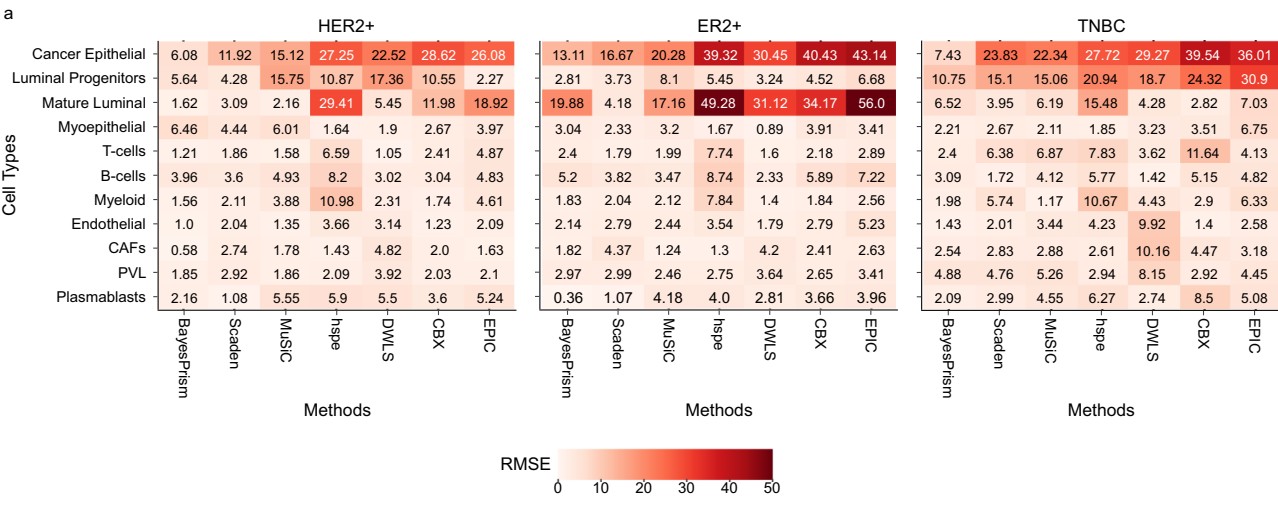

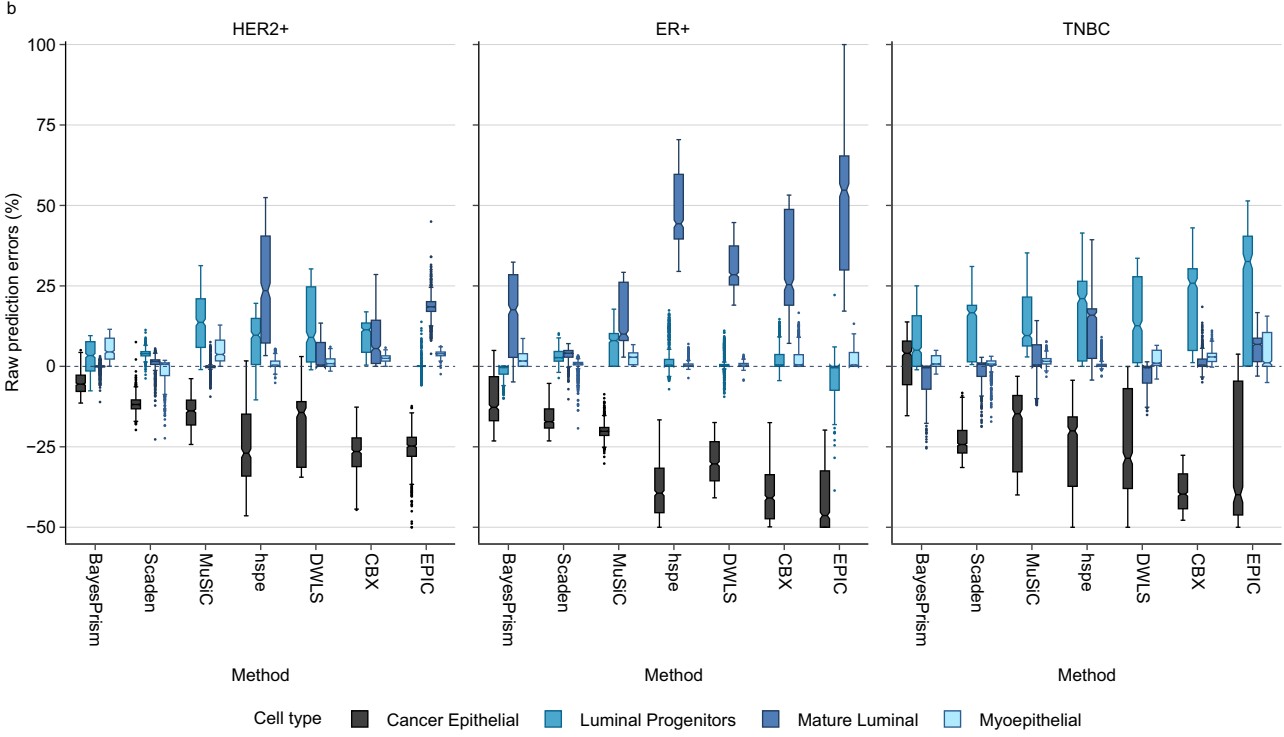

**Fig. 3 | Impact of normal epithelial lineages and molecular subtypes on deconvolution. a** RMSE between predicted and actual cell compositions, aggregated by molecular subtypes (HER2+, ER+ and TNBC). Darker shade of red represents higher RMSE values and poorer performance, with numeric RMSE values shown. Cell types (y-axis) are organised into four categories: cancer epithelial, normal epithelial (luminal progenitors, mature luminal and myoepithelial), immune (T-cells, B-cells and myeloid), and stromal cells (endothelial, CAFs, PVL and plasmablasts). CAFs: Cancer Associated Fibroblasts, PVL: Perivascular-like, RMSE: Root Mean Square Error. **b** Raw prediction errors of seven methods, BayesPrism, Scaden, MuSiC, CBX, DWLS, hspe and EPIC, for cancer epithelial and three minor subtypes of normal epithelial cells aggregated by molecular subtypes (HER2+, ER+ and

TNBC). Higher positive and lower negative raw prediction errors represent poorer performance. Mixtures were synthesised at a fixed purity level of 50% using three minor cell types of normal epithelial cells and eight other major cell types (cancer epithelial, T-cells, B-cells, myeloid, endothelial, CAFs, plasmablasts and PVL). n = 2000 artificial bulk mixtures. Each box represents the middle 50% of raw prediction errors, which includes the first quartile (Q1), the median, and the third quartile (Q3). Upper and lower whiskers depict maxima and minima of raw prediction errors, excluding outliers. Outliers are raw prediction errors that are more than 1.5x the interquartile range from either Q1 or Q3. Zero line indicates a perfect match between prediction and ground truth. Source data are provided as a Source Data file.

expensive and have specific sample processing requirements and thus not amenable to routine clinical practice. Therefore, approaches that accurately estimate cell types within a tissue sample using bulk RNA-seq have been exploited. Using one of the best-annotated breast cancer scRNA-seq dataset to date[15] and three validation datasets[6,37–39], we comprehensively benchmarked the performance of nine TME

deconvolution methods across 19 tumour purity levels, and their ability to estimate three normal epithelial minor cell types in different breast cancer molecular subtypes, and three lineage granularity levels of immune cells. By utilising Bray-Curtis dissimilarity and Aitchison distance, we addressed their compositional nature and enabled mixture-to-mixture comparisons of methods' performance.

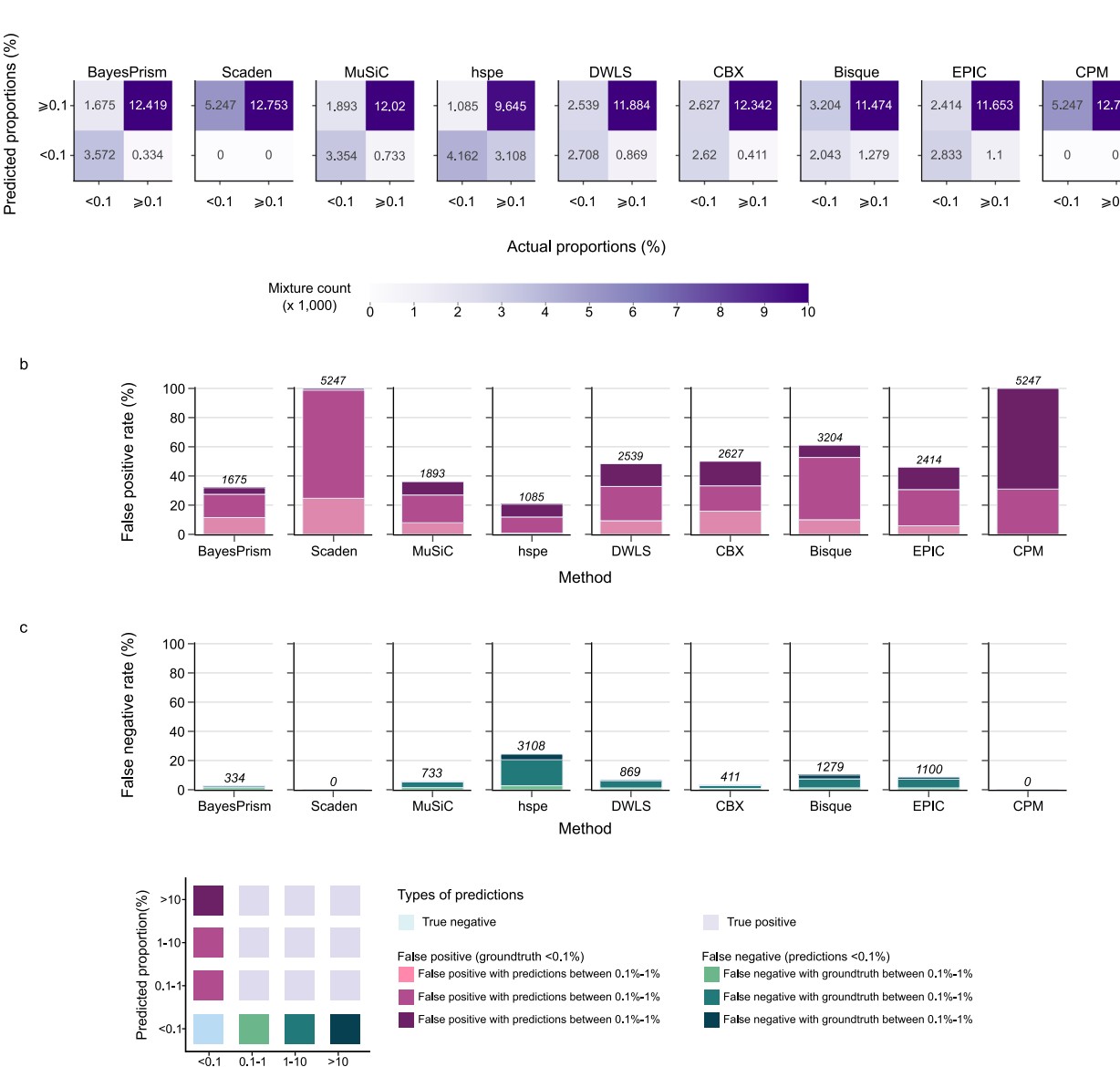

**Fig. 4 | The performance of the nine deconvolution methods assessed by false positive and false negative rates. a** Confusion matrices depicting all nine methods' performance on predicting whether a cell type is absent (< 0.1%) or present (≥ 0.1%) in a mixture. For each confusion matrix, x-axis represents predicted absence/presence, y-axis represents actual absence/presence, and false positive, true positive, false negative, and true negative numbers are aggregated across all cell types. **b** Predicting cell type presence when cell type absent in the mixture. Percentages of the three levels of false positives out of the total number of false positives and true negatives (actual proportion <0.1%). Counts of false positives are shown above each bar for all cell types. **c** Predicting cell type absence when cell type present in the mixture. Percentages of the three levels of false negatives out of total number of all false negatives and true positives (predicted proportions <0.1%). Counts of false negatives are shown above each bar for all cell types. Figure legend for both (**b**) and (**c**) illustrates definitions of true negative, false positive, true positive, and false negative predictions. The more accurate a method in predicting presence/absence, the lower false positive rates and false negative rates are. Source data are provided as a Source Data file.

Obtaining bulk RNA-seq data from thousands of samples with matching cell-type proportions required for benchmarking would be expensive and resource-intensive. Even the use of flow cytometry would not overcome the expense and is technically challenging to apply to large cohorts of solid tumours, preventing the experimental mixing of single cells used in benchmarking studies like ours. Similar to previous deconvolution benchmarking efforts[26–28], we overcome the challenge of data scarcity by mixing data from single cells to create artificial bulk RNA-seq mixtures. Without resampling a single cell more than once for one mixture, proportions of cell types in artificial mixtures are constrained by the number of available single cells in the scRNA-seq data (i.e. constraint sampling). This means a cell type with only 50 cells will never make up more than 10% of any artificial bulk

mixture for a mixture size of 500 cells, and it is more likely that the same set of cells are sampled for different mixtures. To overcome class imbalance problems and sampling individual cells many times, we synthesised new cells using the tool SMOTE[36,42]. On this note, we acknowledge that a potential limitation of SMOTE version we used is it does not attempt to model the underlying distribution of single-cell gene distributions. On the other hand, non-linear and generative deep learning methods such as variational autoencoders[43] and generative adversarial networks[44] can learn the distribution of each class and enforce synthesised samples to fit in such distributions. A notable example is DeepSMOTE[45], which combines the Cartesian distance algorithm of SMOTE with a variational autoencoder architecture. We recommend future studies to explore the potentials of generative

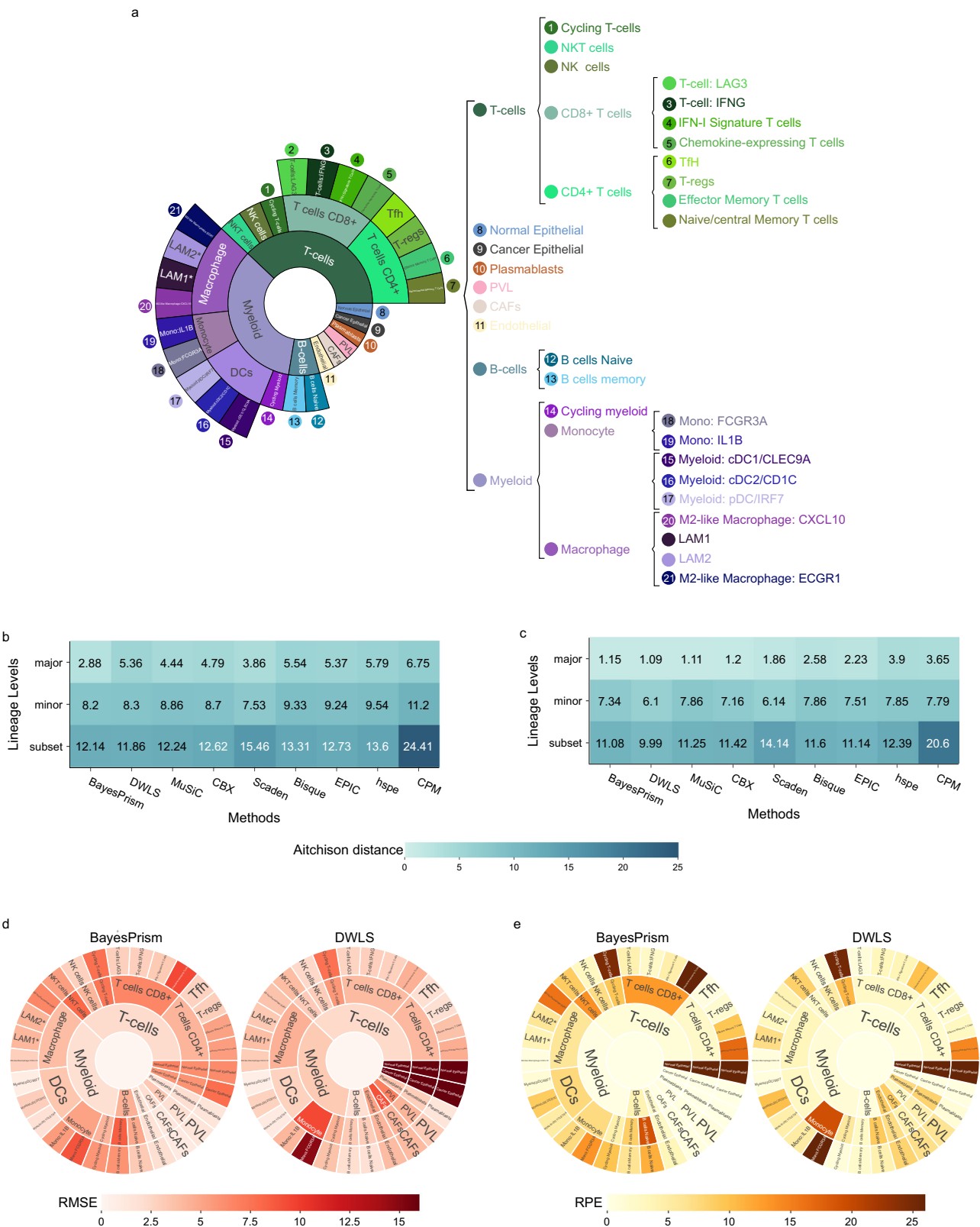

deep learning methods in single-cell data synthesis. For each patient, SMOTE synthesised as many cells as necessary so that cell counts of all cell types were equal to the count of the most abundant cell type, ensuring all cell types are randomly sampled with the same probabilities. We show that this approach synthesised cells that cluster tightly with original cells. SMOTE has been used previously in RNA-seq[46], and while we believe data synthesis is a better alternative to

resampling and constraint sampling, as it introduces more heterogeneity to the artificial mixtures, we do not suggest it as a replacement for real samples. Rather, increasing availability and resolution of single-cell data will be critical for improving performance of deconvolution methods.

In this study, we showed that deconvolution methods may perform worse in the presence of a higher proportion of cancer cells.

**Fig. 5 | Impact of immune lineages on deconvolution. a** The relationship of immune cells in the major, minor and subset cell types. **b, c** Aitchison distance between predicted and actual compositions of 2000 mixtures containing 23 subset cell types of T-cells, B-cells and myeloid at 50% tumour purity level. The median Aitchison distance across 2000 mixtures is shown for each of the nine methods using either **b** all cell types or **c** only immune cell types. Lighter shade of teal indicates smaller Aitchison distance and between performance. RMSE (red) (**d**) and RPE (orange) (**e**) between predicted and actual cell proportions of BayesPrism and DWLS, aggregated into major, minor and subset cell types. Darker shades of red

and orange represent higher RMSE and RPE values and poorer performance, respectively. Cancer epithelial, normal epithelial, endothelial, CAFs, PVL and plasmablast cell types were used for artificial bulk simulation at all three levels and, therefore, possess three sets of RMSE and RPE values across the lineage levels. Several minor immune cell types, such as NK cells or memory B-cells, do not have any subset cell types and were therefore re-used at the subset level, resulting in two sets of RMSE and RPE values at minor and subset level. RMSE: Roost Mean Square Error, RPE: relative proportion error. Source data are provided as a Source Data file.

Newman et al.[18] showed that Pearson's r values of CIBERSORT, an earlier version of CIBERSORTx, and 6 other deconvolution methods[47–51] declined as tumour purity levels in simulated mixtures increased. We confirm this pattern for the newer deconvolution methods, and in addition found that BayesPrism, Scaden, MuSiC, CBX and DWLS are superior compared to Bisque, hspe, EPIC, and CPM in overall performance and in deconvolution of major immune cell types across different tumour purities. In addition, we show the cause of the performance decline in increasing tumour purity is due to methods mistaking normal epithelial for cancer epithelial cells in high tumour-purity mixtures. Similar patterns were also observed when we used simulated mixtures generated from two additional scRNA-seq datasets, Bassez et al.[37] and Pal et al.[38], indicating the robustness of our findings when technical batch effect is present. Furthermore, our validation work on Bassez et al.[37] and Pal et al.[38] confirmed BayesPrism, MuSiC and Scaden are resilient against variable tumour purity, and their good performance on deconvolving major immune cell types is generalisable across datasets. In Pal et al.[38], where normal epithelial cell labels were provided, we also confirmed the normal-cancer mis-prediction for MuSiC, CBX, DWLS, and hspe.

The true test for deconvolution methods is how they perform on real bulk mixtures. Our validation analysis on 1038 breast cancer samples from TCGA revealed that the three methods showing highest level of resilience against tumour purity (BayesPrism, Scaden and MuSiC) were also the top-performing methods in predicting tumour purity, compared against Aran et al.[40] estimates, and tumour-infiltrating lymphocyte (TIL) content, compared against Saltz et al.[41] estimates. CBX and hspe also showed comparable performance to BayesPrism, Scaden, and MuSiC in predicting tumour purity. We do, however, note that most methods only achieved sub-optimal performance with TIL estimation, which could have been a result of the RNA-seq extraction sample and the H&E image used for true TIL content having been collected from two tumour regions.

We also found that cancer cells were mis-predicted as different subtypes of normal epithelial cells depending on the breast cancer subtype, observed in the predictions BayesPrism, Scaden, MuSiC, CBX, DWLS, hspe and EPIC. This observation supports that there are transcriptional similarities between normal epithelial cell subtypes and different cancer subtypes, which could be reflective of the cancer cell of origin. Luminal progenitor cells were generally overpredicted as TNBC cancer samples, aligning with the hypothesis that basal-like cancers, which overlap with TNBC subtype[52], originate from luminal progenitor cells[53,54]. Mature luminal cells were generally overpredicted for ER+ cancer samples. These cells are hormone receptor positive and are reported to have molecular profiles closest to luminal A cancers[53], which are generally ER+/PR+[52], although the cell of origin for this subtype is yet to be definitively confirmed[54]. Finally, HER2+ cancers had a less definitive normal cell type mis-predictions, which varied depending on the deconvolution method, although mostly reflected luminal cell lineage rather than myoepithelial lineage. This aligned with the hypothesis that HER2+ cancers arise through HER2 (ERBB2) amplification in cells committed to the luminal lineage[53,54].

Granularity is a recent focus in deconvolution research, with several of the latest methods dedicated to estimating populations of rare immune subtypes such as MuSiC[20] or DWLS[23]. Prior to the advent

of these methods, Newman et al.[18] and Jiménez-Sánchez et al.[27] suggested that deconvolution performance is not uniform across the immune subtypes. In this study, we conducted cross-lineage-level comparisons and showed that deconvolution performance decreased from major to minor to subset immune cell types. At subset level, even the two best-performing methods, BayesPrism and DWLS, did not produce optimal predictions and had numbers of false-positive and false-negative failures for all cell types. Interestingly, the second best-performing method DWLS was purposely built to deconvolve rare cell types, by dynamically penalising more abundant cell types and weighting up rare cell types based on their population in the single-cell reference matrix. Increasing availability of scRNA-seq data with larger representation of rare cell types will likely help address this issue.

We acknowledge potential limitations of our study. First, we did not assess enrichment-based methods such as ConsensusTME[27], TIMER 2.0[28], and xCell[55], which use single-sample Gene Set Enrichment Analysis (ssGSEA)[56,57] to calculate collective up-or down-regulation of known marker genes of specific immune cell types (enrichment scores). Mixtures with higher populations of an immune cell type will have higher enrichment scores using its related marker genes. This is a more objective metric compared to the relative nature of cell proportions. In addition, ConsensusTME[27], TIMER 2.0[28], and xCell[55] have each incorporated marker gene signatures from multiple cancer types for pan-cancer deconvolution, which currently no fraction-based methods (either single-cell-based or deep learning methods) achieve. However, we did not include these methods in this study, as it was not feasible to assess them together with the proportion-based deconvolution methods, and based on the experiments and metrics used herein. The second potential limitation in our study is the process of dissociating tumours and the microfluidics kinetics, used in the 10X Genomics Drop-Seq for single-cell capture, can potentially lead to an overrepresentation of immune cells. In addition, there are granulocytes such as Neutrophils that are not captured with Drop-Seq. For future studies, single-cell spatial transcriptomics is likely to be the solution for this issue. Lastly, in this study we fixed cell count of all pseudobulk mixtures at 500. This was mainly to ensure that we did not resample the same cell twice for a mixture, as some samples only contained between 500 and 1000 for the most prevalent cell type. With the average number of reads per cell of approximately 7000 in Wu et al.[15], a 500-cell mixture would have around 3.5 million reads per pseudobulk, which is less than the typical amount of 25–50 million reads. This small library size could have led to underrepresentation of lowly expressed genes compared to real bulk data.

Overall, we found that BayesPrism, MuSiC, DWLS, CBX and Scaden are the most robust deconvolution methods against changing biological conditions of the TME. Four of these approaches, BayesPrism, MuSiC, DWLS and Scaden, also showed the best performance using bulk RNA-seq to predict tumour purity. While all four methods were robust against variable tumour purity levels, BayesPrism and DWLS showed the most resilience with higher cell type granularity. The increasing availability of more diverse and well-annotated scRNA-seq datasets will greatly facilitate deconvolution of rare immune cell types. For improved deconvolution of rare immune subtypes, we expect future methods to build upon purposely developed algorithms such as the dynamic weighting system in DWLS and the hierarchical

proportioning system in MuSiC. We also anticipate deep learning methods to become more popular and translate the success of incorporating biological knowledge using Graph Convolutional Neural Networks (GCNN) from domains of cancer research to deconvolution[58–60]. Deep learning methods can also be useful in pan-cancer deconvolution for fraction-based methods. The prospect of a deep learning method trained on simulated mixtures from many cancer types is interesting and should be explored.

## Methods

### Ethics approval and consent to participate

This project used scRNA-seq and bulk RNA-seq data from breast cancer patients which was previously published. The QIMR Berghofer Human Research Ethics Committee approved use of public data (P2095).

### Dataset selection and pre-processing

In this study, we collected raw gene counts and annotated single cell labels, including cancer cell labels which were inferred using Copy Number Variations (CNV), from the breast cancer scRNA-seq datasets in Wu et al.[15], Bassez et al.[37], and Pal et al.[38]. For all 3 datasets, we normalised downloaded raw gene counts by counts-per-10,000 using Seurat v3 NormalizeData() function with normalisation method set to relative counts (RC), which applied counts-per-10,000 normalisation without log transformation. The resulting normalised counts were then used for downstream benchmarking analyses.

Of note, we were not able to use CPM for Bassez et al.[37] and Pal et al.[38], as the CPM approach requires UMAP coordinates which were not available for these additional datasets.

**Wu et al.** The annotated scRNA-seq data from 26 patients representing 3 major clinical subtypes with 100,064 single cells was accessed from Gene Expressions Omnibus (GEO, GSE176078). The downloaded data were mapped to GRCh38 human reference genome and included filtered Unique Molecular Identifier (UMI) gene counts (excluding cells with gene and UMI counts below 200 and 250, respectively, and mitochondrial percentages less than 20%), UMAP coordinates and cell annotations. The cell annotation included nine major cell types, 29 minor cell types, and 49 subset cell types.

**Bassez et al.** The annotated scRNA-seq data from Bassez et al.[37] was accessed via the study's official website at https://lambrechtslab.sites.vib.be/en/single-cell. The study includes 2 patient cohorts: 39 patients who received one dose of pembrolizumab before surgery (cohort 1) and 15 patients undoing neoadjuvant chemotherapy before receiving pembrolizumab and the subsequent surgery (cohort 2), with single-cell sequencing conducted both pre-and post-treatment. We only used pre-treatment cell counts and cell annotations of 42 patients (31 from Cohort A and 11 from Cohort B) from whom scRNA-seq data was retrievable, which include 105,222 single cells representing 3 major clinical subtypes. The downloaded gene counts were mapped to GRCh38 human reference genome and excluded cells expressing <200 or >6000 genes, cells containing less than 400 UMIs, as well as cells with more than 15% mitochondrial reads. Cell annotations included eight major cell types.

**Pal et al.** We downloaded scRNA-seq profiles of 52 patients in Pal et al.[38] from GEO (GSE161529), which includes tumour samples as well as their matching normal and pre-neoplastic samples. Following the download analysis guide in[61], we retrieved cell annotations of from Seurat-integrated objects of 13 ER+, 6 HER2+ and 8 triple-negative tumour samples, comprising of 148,694 single cells from 27 patients. The downloaded UMI counts excluded cells with less than 500 genes or less than 20% mitochondrial reads, as well as cells with unusually high number of reads or genes. Cell annotations included 12 major cell types.

### Oversampling of minority cell types

The data synthesis method Synthetic Minority Oversampling Technique (SMOTE)[36] was used to generate new cells for under-represented classes before using single-cell RNA-seq data to simulate artificial bulk mixtures. SMOTE is a data augmentation technique that synthesises new data based on the original data points. For each data point in a cell type, SMOTE draws a vector between the point and a random point of the same cell type before creating a new data point at a random place along this vector. The result is new data points that are not exact replicates of, but also not too deviated from the original data. The Python implementation of Distance SMOTE[42] v0.4.0 (https://github.com/analyticalmindsltd/smote_variants) was applied to cells for each individual patient sample. To ensure synthesised cells are representative of original cells, for experiment, we discarded cell types with less than 10 cells before executing SMOTE. After data oversampling, Seurat v3.5 implementation of UMAP was executed on the output gene counts.

To preserve inter-patient variability, we conducted SMOTE separately per patient, i.e. synthesized cells are created for each patient using only the patient's original cells. This means if a cell type is not present for a patient, its cell count after SMOTE is still 0.

Additionally, cohort 1 and cohort 2 in Bassez et al.[37] were sequenced separately, resulting in different gene lists in their scRNA-seq profiles (25,288 genes for Cohort 1 and 22,889 genes for Cohort 2). After counts-per-10,000 normalisation, we used all genes for each patient in Bassez et al.[37] during SMOTE and only retained the 22,567 intersecting genes between two cohorts for the subsequent generation of single-cell reference profiles and simulated bulk mixtures. Unique gene list of each cohort and their intersecting gene list are provided in Supplementary Data 4.

### Splitting data into training and test

The deconvolution methods benchmarked in this study require one of three types of training input: 1) a single-cell reference profile matrix containing gene expression of all cells from the training dataset (Bisque, CBX, CPM, DWLS, hspe, and MuSiC), 2) a signature matrix containing differentially expressed genes for all cell types (EPIC), or 3) simulated bulk RNA-seq mixtures with known cell fractions for training a model (Scaden). In addition, all methods require separate test data for performance evaluation. As methods must be completely blind to the test data, we assigned all cells from certain patients to training data and those from the remaining patients to the test data, ensuring all cell types and molecular subtypes are present in both train and test data. Supplementary Fig. 1 details which patient was assigned to train/test and cell-type-specific counts at each linear level for Wu et al.[15].

### Simulation of training artificial bulk RNA-seq mixtures

To generate bulk RNA-seq mixtures with known cell fractions, cells were randomly sampled from the training dataset to create artificial bulk mixtures. This random sampling was constrained to cells from only one patient per mixture to capture inter-patient gene expression heterogeneity. The code implementation was based on the Python packages pandas v1.1.5 and numpy v1.19.15 and was influenced by a similar procedure used in Menden et al.[17]. First, the total number of cells per mixture was fixed to 500 cells. For each mixture, a random number of cell types between 5 and maximum available cell types for a patient was chosen to be included. Then, a random fraction between 0 and 1 was generated and assigned to a random cell type, which effectively enables one random cell type per mixture to have a free-range proportion:

$$f_c = random.default_{rng}().choice() \qquad (1)$$

where $f_c$ is the free-range proportion of cell type $C$. We then randomly generated fractions for other cell types, normalised them between [0,

1] and then normalised them again between $[0, 1\text{-}f_c]$ to ensure cell-type proportions sum to 1:

$$r_{\bar{c}} = random.default_{rng}().choice() \qquad (2)$$

$$f_{\bar{c}} = \frac{r_{\bar{c}}}{\sum_{\bar{c}_{all}} \bar{c}} \times (1 - f_c) \qquad (3)$$

where $r_{\bar{c}}$ and $f_{\bar{c}}$ are non-normalised and normalised proportions, respectively, for any cell type besides $C$. Then each fraction was multiplied by 500 to retrieve the corresponding cell counts for each cell type:

$$n_c = f_c \times 500 \qquad (4)$$

$$n_{\bar{c}} = f_{\bar{c}} \times 500 \qquad (5)$$

where $n_c$ is cell counts of cell type $C$, $n_{\bar{c}}$ is cells counts of any other cell type.

For each simulated bulk mixture, we repeated this process to collect a unique set of single cells and sum counts-per-10,000 normalised counts across all 500 cells to produce the simulated bulk counts. We used this process to generate different sets of train mixtures with known cell-type proportions, which were used as training data for the deep-learning model Scaden.

**Variable tumour-purity mixtures.** We used labels of major cell types from Wu et al.[15] ($n = 9$), Bassez et al.[37] ($n = 8$), and Pal et al.[38] ($n = 11$) to randomise cell-type-specific proportions for each simulated mixture. We generated 5000 mixtures per patient, resulting in 90,000 mixtures per experiment from Wu et al.[15], 110,000 mixtures per experiment from Bassez et al.[37], and 80,000 mixtures per experiment from Pal et al.[38]. Each mixture was simulated using single cells from only one scRNA-seq dataset and bulk deconvolution only relied on single-cell reference profiles from the same dataset.

**Normal epithelial lineages mixtures.** We used single cells from Wu et al.[15] for this experiment as it was the only dataset with lineage of normal epithelial cells. We used major cell type labels for cancer epithelial, T-cells, B-cells, myeloid, endothelial, cancer-associated fibroblasts (CAFs), perivascular-like cells (PVL), and plasmablasts, and replaced normal epithelial with their subtype labels (luminal progenitors, mature luminal, and myoepithelial) (Supplementary Data 2). We generated two different sets of train mixtures, one where all three minor cell types of normal epithelial were included in the simulation process and another where they were all excluded. Each set had 5,000 mixtures per train patient, resulting in 90,000 mixtures in total.

**Immune lineages mixtures.** We used single cells from Wu et al.[15] for this experiment as it was the only dataset with lineages of immune cell types at both minor and subset levels. We generated two sets of artificial bulk mixtures using minor and subset immune cell annotations. In the two datasets, we used major cell type labels for cancer epithelial, normal epithelial, endothelial, CAFs, PVL, and plasmablasts. We then used the 11 minor cell-type annotations of T-cells, B-cells, and myeloid cells, and generated 5000 mixtures per patient (90,000 mixtures in total) (Supplementary Data 2). This formed our train data at minor immune lineage level. In addition, we also used 17 subset cell-type and six minor cell-type annotations (which do not have any subset cell types) of T-cells, B-cells and myeloid to generate another set of 5000 mixtures per patient (90,000 mixtures in total) (Supplementary Data 2). This formed our train data at subset immune lineage level. Note that at subset level, we did not use T-cells:GZMK, M2-like

Macrophage, and Myeloid:DC/LAMP3, as they were only detected in one train set patient and MuSiC is a multi-subset method.

**Generation of single-cell reference profiles and signature matrix**
BayesPrism, Bisque, CBX, CPM, DWLS, hspe, and MuSiC require a single-cell reference matrix containing all cell types present in the deconvolved bulk mixtures. For each set of simulated train mixtures described above, we generated the corresponding single-cell reference profiles by discarding oversampled cells and retaining scRNA-seq data from only original cells in patients selected for training. For EPIC, we provided it with the signature matrix produced by CBX. This means patient-specific and cell-type counts detailed in Supplementary Data 2 also depicts the compositions of each single-cell reference profile.

**Simulation of test artificial bulk RNA-seq mixtures**
The simulation process used to generate artificial training mixtures was adapted to generate different sets of test mixtures.

**Variable tumour-purity mixtures.** We used the same major cell-labels from Wu et al.[15] ($n = 9$), Bassez et al.[37] ($n = 8$), and Pal et al.[38] ($n = 11$) as were used in the training mixture simulation. However, tumour purity level of each mixture was fixed at one of the 19 values between 5% and 95% with 5% interval. All other cell types were randomised similarly to the training simulation. We generated 250 mixtures per purity level per test patient sample, resulting in a total of 38,000 mixtures from Wu et al.[15], 57,000 mixtures from Bassez et al.[37], and 38,000 mixtures from Pal et al.[38].

For the technical batch effect validation experiment where single-cell reference and train simulated mixtures come from Wu et al.[15], we grouped original cell-type labels in Bassez et al.[37] and Pal et al.[38] consistently with cell-type labels from Wu et al.[15]. For Pal et al.[38], we grouped tumour-associated macrophages (TAMs) and dendritic cells (DCs) into myeloid. We also dropped pericytes, as they are not annotated in Wu et al.[15]. This resulted in 8 major cell types: cancer epithelial, normal epithelial, T-cell, B-cell, myeloid, endothelial, CAFs, and plasmablasts. For Bassez et al.[37], we grouped mast cells and plasmacytoid dendritic cells (pDC) into myeloid, resulting in 6 major cell types: cancer epithelial, T-cell, B-cell, myeloid, endothelial, and CAFs. After this cell annotation grouping step, 57,000 mixtures from Bassez et al.[37], and 38,000 mixtures from Pal et al.[38] were generated using the same process described above. For each set of simulated mixtures, we only retained the intersecting genes between Bassez et al.[37] and Wu et al.[15], and Pal et al.[38] and Wu et al.[15], respectively (Supplementary Data 5).

Similar to train mixture simulation, each test mixture was simulated using single cells from only one scRNA-seq dataset.

**Normal epithelial lineages mixtures.** We used the same minor cell-type labels of normal epithelial cells and major cell-type labels for other cell types from Wu et al.[15] as were used in the train mixture simulation. Also similar to train mixture simulation, we generated two different sets of test mixtures, one where all three minor cell types of normal epithelial were included in the simulation process and another where they were all excluded. Both sets had 250 mixtures per test patient (2000 mixtures in total, Supplementary Data 2).

**Immune lineages mixtures.** We used the same minor and subset cell-type labels of immune cell types and major cell-type labels for other cell types from Wu et al.[15] as were used in the train mixture simulation. Also similar to train mixture simulation, we generated two different sets of test mixtures, one with minor and one with subset immune cell types. Both sets had 250 mixtures per test patient (2000 mixtures in total, Supplementary Data 2).

## Validation on real bulk mixtures from TCGA breast cancer samples

RNA-seq data from the TCGA breast cancer project was obtained from the UCSC Cancer Genomics Hub (no longer operational) in November 2014. Reads were trimmed for adapter sequences using Cutadapt[62] (version 1.9) and aligned using STAR[63] (version 2.5.2a) to GRCh38 human reference genome with the gene, transcript, and exon features of Ensembl (release 84) gene model. Quality control metrics were computed using RNA-SeQC[64] (version 1.1.8) and expression was estimated using RSEM[65] (version 1.2.30). The analysis was restricted to 1038 primary tumour samples with intragenic rate >95% and protein-coding rate >90%. Gene counts were normalised by Transcript-per-Million (TPM).

To run deconvolution on TCGA bulk mixtures, we reused the same single-cell reference profiles (for Bisque, BayesPrism, CBX, CPM, DWLS, hspe, and MuSiC) and train simulated mixtures (for Scaden) with annotations of nine major cell types from Wu et al.[15] that were generated for the variable tumour purity experiment. We also provided EPIC with the signature matrix generated by CBX.

For PAM50 subtyping, lowly expressed genes were filtered, followed by upper-quantile normalisation and log RPKM value transformation using edgeR[66] package. PAM50 subtyping was performed using genefu[67] package.

**Validation of tumour purity estimations.** We downloaded the Consensus Tumour Estimates (CPE) of 1113 breast cancer patients in TCGA from Aran et al.[40], which was unified across tumour purity estimates by ABSOLUTE[68], ESTIMATE[69] and LUMP[40], as well as pathologist-annotated estimates (Pathology)[40] (Supplementary Fig. 16). Out of these 1113 patients, we were able to match 1031 patients back to the downloaded transcriptomics data using patient barcodes. We validated deconvolution performance on purity by comparing predicted cancer epithelial proportions of populations against CPE tumour purity estimations from the filtered list of 968 patients (Supplementary Data 6).

**Validation of lymphocyte estimations.** We downloaded tumor-infiltrating lymphocytes (TIL) estimations of 944 breast cancer patient patients in TCGA from Saltz et al.[41], which was produced by a trained deep-learning model using H&E images. This deep learning model was trained using H&E images manually annotated by pathologists and shown to have strong correlation with cellular compositions derived by the deconvolution method CIBERSORT[18], an earlier iteration of CIBERSORTx. Out of 944 patients, we were able to match 892 patients back to the downloaded transcriptomics data using patient barcodes. We validated deconvolution performance on lymphocyte populations by comparing total predicted proportions of T-cells and B-cells against TIL estimations from Saltz et al. from the filtered list of 892 patients (Supplementary Data 7).

## False positives and false negatives in tumour deconvolution

To analyse false positive rates, we identified true negatives, here classed as actual cell type proportion <0.1%, and binned the predicted proportions into four intervals (<0.1%, 0.1%−1%, 1%−10% and >10%). Predicted proportions <0.1% were considered true negatives (TN), i.e. correctly predicting a cell type is missing. Predictions in the other three intervals were considered false positives (FP). We counted the number of occurrences of each interval and calculated false positive rates as FP/(FP + TN).

To analyse false negative rates, we identified true positives (actual cell type proportions ≥0.1%) and binned the actual proportions into three intervals (0.1%−1%, 1%−10% and >10%). We also identified predicted negatives (predicted cell type proportions of <0.1%). Actual proportions ≥0.1% were true positives (TP). Predictions of <0.1% in the three intervals were considered false negatives (FN). We then counted the number of occurrences of each interval and calculated false negative rates as FN/(FN + TP).

## Overview of TME deconvolution methods

In this study, we benchmarked three categories of deconvolution methods: deep learning (Scaden[17] v1.1.2), single-cell-referenced-based (BayesPrism[29] v2.0, Bisque[22] v1.0.5, CIBERSORTx[19] – available as of 29th June 2021, CPM[24] v0.1.6, DWLS[23] v0.1.0, hspe[35] v0.1, MuSiC[20] – available as of 29th June 2021) and signature-based (EPIC[25] v1.1). When further data analysis was required, we employed the R package Seurat v3.5, as well as the Python package Distance SMOTE v0.4.0, pandas v1.1.5, numpy v1.19.15, scanpy v1.7.2, scikit-learn v0.24.2, and scikit-bio v0.5.6.

Of note, the methods BayesPrism, Bisque, MuSiC and hspe could incorporate cell-type-specific marker genes during deconvolution. By contrast, DWLS could either use Seurat or MAST to build its internal signature matrix. In this study, we chose to use default parameters for all methods and hence reported results produced by the no-marker-genes version of BayesPrism, Bisque, MuSiC and hspe, and the Seurat version of DWLS. We have, however, provided a performance comparison across all three immune lineage levels in Supplementary Fig. 18, which shows slight differences for BayesPrism, Bisque, MuSiC and better performance for hspe when marker genes are not used (Supplementary Fig. 18a, b), as well as slightly better performance for DWLS-with-Seurat (Supplementary Fig. 18c, d). When performance optimisation is of concern in future studies, we do recommend considering different parameter settings for each method.

Here we provide a description of how these methods were executed in this study. Summarised technical overview of the methods is provided in Supplementary Note 1.

**BayesPrism.** BayesPrism v2.0 was installed from the GitHub repository https://github.com/Danko-Lab/BayesPrism. In addition to cell types, BayesPrism allows the option to specify cell subtypes, known as cell states, in the single-cell reference matrix. Deconvolved fractions of cell states of each cell type would be summed to produce the cell type's fraction. To make BayesPrism comparable with other methods, we chose not to use this option and only specify cell types for BayesPrism. All other parameters were set as the algorithm's default options. Lastly, BayesPrism produces two sets of predicted cell-type fractions, one before and one after the Gibbs sampling step, in which the reference matrix is updated using within-sample tumour expression and across-samples non-tumour expression. We used post-Gibbs-sampling results for BayesPrism in this study and have provided a cross-dataset performance comparison of pre- versus post-Gibbs-sampling in Supplementary Fig. 17.

**bisqueRNA (Bisque).** BisqueRNA (Bisque) v1.0.5 was installed from https://cran.r-project.org/web/packages/BisqueRNA. In Jew et al.[22], single-cell gene counts were normalised, filtered for variable genes, and scaled using the SCTransform function. We chose not to filter raw counts for variable genes to ensure identical gene sets in reference single-cell data across all methods. Moreover, the benchmarking study by Cobos et al.[26] suggested that performance of Bisque remains relatively unchanged across many normalisation methods. For these reasons, we chose not to process single-cell data for Bisque using the SCTransform function, but instead kept gene counts linear, similarly to other methods. All other parameters were set as the algorithm's default options.

**CIBERSORTx (CBX).** The containerised version of CIBERSORTx (CBX) was downloaded from https://hub.docker.com/r/cibersortx/fractions. We executed CBX using the algorithm's default parameters. CBX also offers several functionalities to deconvolve tumours across sequencing platforms and infer cell-type-specific gene expression values, among which was the S-mode option used for batch effect correction

between bulk mixtures and droplet-based single-cell reference. We utilised this functionality when running deconvolution methods on TCGA bulk samples.

**Cell Populating Mapping (CPM).** Cell Population Mapping (CPM) v0.1.6 was installed from the CRAN repository https://cran.r-project.org/web/packages/scBio. When running CPM, all parameters were set as the algorithm's default options, apart from cell-state space and neighbourhood size. As suggested in[24], we chose to employ UMAP coordinates as the cell-state space. This was achieved by obtaining the original UMAP coordinates of train set patient samples in Wu et al.[15].

CPM instructions recommend choosing a neighborhoodSize value representing the maximum number of cells of the same cell type surrounding any random cell in the cell-state space. As our cell-state space was a dense UMAP distribution, we aimed to select the highest possible value for neighborhoodSize. We followed CPM's tutorial where neighborhoodSize is roughly one-fourth of the count of the smallest cell type. As our least abundant cell type was 1330 (normal epithelial cells), we chose neighborhoodSize=250.

**Dampened Weighted Least Squares (DWLS).** Dampened Weighted Least Squares (DWLS) v0.1.0 was installed from the CRAN repository https://cran.r-project.org/web/packages/DWLS. All parameters used for running DWLS were set as the algorithm's default options.

**Estimating the Proportion of Immune and Cancer cells (EPIC).** Estimating the Proportion of Immune and Cancer cells (EPIC) v1.1 was installed from the GitHub repository https://github.com/GfellerLab/EPIC. Unlike deep learning methods or single-cell-based methods, EPIC is a signature-based method and requires a matrix of expression values of differentially expressed genes for all present cell types. We could not achieve this using EPIC's default signature matrix as it does not include all cell types in our scRNA-seq data. To overcome this, we chose to run EPIC using the signature matrix that CBX produced for each corresponding experiment. All other parameters were set as the algorithm's default options.

**hybrid-scale proportions estimation (hspe).** Hybrid-scale proportions estimation (hspe) v.01 was installed from the Github repository https://github.com/gjhunt/hspe. All parameters used for running hspe were set as the algorithm's default options. Importantly, hspe assumes input gene counts of both single-cell reference profiles and bulk mixtures are in logarithmic space and applies an inverse-logarithm step before deconvolution to convert counts to linear space. Due to this reason, we applied *log2(gene_counts+1)* transformation for hspe.

**MUlti-Subject SIngle Cell deconvolution (MuSiC).** MUlti-Subject SIngle Cell deconvolution (MuSiC) was installed from the GitHub repository https://github.com/xuranw/MuSiC, using the version made available on 29th June 2021. Notably, MuSiC implements a multi-subject gene-weighting approach, which dynamically penalises genes with low between-subject variance for the single-cell reference profiles. This feature is a core step of MuSiC deconvolution process, and we therefore chose to utilise it by including matching patient IDs for the single-cell reference. Additionally, MuSiC can optionally deconvolve cell subtypes before adding fractions of subtypes together to form cell type fractions. For MuSiC to be comparable with other methods, we chose not to use this functionality. All other parameters used to run MuSiC were set as the algorithm's default options.

**Single cell–assisted deconvolutional DNN (Scaden).** Scaden v.1.1.2 was installed from the Bioconda repository https://anaconda.org/bioconda/scaden. The simulated train mixtures were used to train the Scaden model for 15,000 steps for each experiment. The trained

model was then used to predict compositions of the test mixtures for each experiment.

**Evaluation metrics**
**Compositional evaluation metric.** Cell-type proportions are compositional, i.e. the total sum of all cell types within one mixture is always 100%, and a change in one cell type influences other cell types. We used two compositional evaluation metrics, Bray-Curtis dissimilarity and Aitchison distance[70], to assess the overall mixture-to-mixture deconvolution performance. These metrics model a mixtures as multi-dimensional data point and each cell type within a mixture as a dimension, e.g. a mixture consisting of nine major cell types is considered a nine-dimensional data point. Under this paradigm, prediction errors are how close or similar the predicted multi-dimensional data points are to the true data points.

We used Bray-Curtis dissimilarity to assess influence of variable purity levels on deconvolution, which is measured as:

$$BrayCurtis = 1 - \frac{2C_{pred/truth}}{S_{pred} + S_{truth}} \tag{6}$$

where $C_{pred/truth}$ is the total sum of lesser counts for each cell types in both predicted and true mixture, $S_{pred}$ is the total number of cells in the predicted mixture, and $S_{truth}$ is the total number of cells in the true mixture. Of note, this formula implies that Bray-Curtis dissimilarity is a count-based metric rather than a percentage-metric. However, we could use it in this study as cell counts for all mixtures were fixed at 500, essentially rendering cell counts and cell percentages equivalent.

We used Aitchison distance to assess deconvolution performance across different lineage levels of immune cell types (major, minor, and subset). As recommended by Aitchison et al.[70], we applied Centred Log-Ratio (CLR) transformation on proportions and calculated the Aitchison distance. Prediction errors are the Aitchison distance between a mixture's ground truth and prediction data points:

$$AitcsD = \sqrt{\sum_{i=1}^{k} (CLR(Pred_i) - CLR(Truth_i))^2} \tag{7}$$

where i is each cell type and $k$ is the number of cell types for each mixture.

**Cell-type specific evaluation metric.** We used Root Mean Square Error (RMSE) to measure pair-wise margin of error and Pearson's r to measure pair-wise correlation between predicted and actual cell proportions. These values are computed differently across all cell types compared to each cell type. When evaluating performance for each cell type, we computed RMSE and Pearson's r across all mixtures and presented the median value for each metric. We relied primarily on RMSE over Pearson's r as the pair-wise metric when ranking methods' performance, as higher Pearson's r values can occur to predictions with both high and low RMSE values[26,71].

To investigate the direction of mispredictions, we used raw prediction errors (predicted proportion – actual proportion). Of note, we kept all raw prediction error values as is and only used them for distribution plots (such as boxplots), as averaging or adding them would cause positive and negative values to eliminate each other.

Rare cell types typically have low proportions, which makes the relative magnitudes between the margin of error and the ground truth can be of great importance. For example, a margin of error of 5% to a ground truth of 2% (for a rare cell type) is much more severe than to a ground truth of 30% (for a non-rare cell type). This problem usually manifests in artificially low values of RMSE for rare cell types. Furthermore, this also makes comparison of deconvolution methods between mixtures with a lot of rare cell types and mixtures with only a few cell types challenging. To address this, we calculated Relative

Proportion Error (RPE) for each cell type $i$ in each mixture $j$ as follows:

$$RPE_{ij} = \frac{|Pred_{ij} - Truth_{ij}|}{Truth_{ij}} \qquad (8)$$

## Reporting summary

Further information on research design is available in the Nature Portfolio Reporting Summary linked to this article.

## Data availability

This project used previously published scRNA-seq data available at GSE176078 from Wu et al.[15], count data from [https://lambrechtslab.sites.vib.be/en/single-cell] for Bassez et al.[37] (raw data for this data is available at [https://ega-archive.org/studies/EGAS00001004809]), and GSE161529 from Pal et al.[38]. We accessed count level data that can be accessed publicly. RNA-seq data from TCGA for breast cancer was downloaded from UCSC Cancer Genomics Hub, currently available at [https://portal.gdc.cancer.gov]. Source data are provided with this paper.

## Code availability

The R and Python used for analyses in this study is available at https://github.com/MedicalGenomicsLab/deconvolution_benchmarking[72].

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

## Acknowledgements

We are grateful to the following grants National Health and Medical Research Council of Australia (NHMRC) Senior Research Fellowship (APP1139071) and Investigator Grant (2018244) to N.W. and NHMRC Emerging Leader 1 Investigator Grant (APP2008631) and Metro North Collaborative Research Grant (CRG-179-2020) to O.K. This work and this research were performed on QIMR Berghofer computing infrastructure supported by The Ian Potter Foundation and The John Thomas Wilson Endowment. EDW is based at the Translational Research Institute (TRI), which is supported by a grant from the Australian Government and funding from the Queensland Government; The Atlantic Philanthropies; University of Queensland; Queensland University of Technology; and Mater Research. K.A.T. was the recipient of the Maureen and Barry Stevenson PhD Scholarship, we are grateful to Maureen Stevenson for her support. The results shown here are in part based upon data generated by the TCGA Research Network: https://www.cancer.gov/tcga.

## Author contributions

K.A.T.: Conceptualisation, Methodology, Software, Formal analysis, Investigation, Writing – Original Draft, Writing - Review & Editing, Visualisation. V.A.: Validation, Writing - Review & Editing. D.L.:

Conceptualisation, Writing - Review & Editing. A.B.: Conceptualisation, Writing - Review & Editing. R.J.: Data Curation, Writing - Review & Editing. L.T.K.: Data Curation, Writing - Review & Editing. S.W.: Resources, Writing - Review & Editing. S.Z.W.: Data Curation, Writing - Review & Editing. D.R.: Data Curation, Writing - Review & Editing. G.Al-E.: Data Curation, Writing - Review & Editing. A.S.: Data Curation, Writing - Review & Editing. E.D.W.: Writing - Review & Editing, Supervision. J.V.P.: Resources, Writing - Review & Editing, Supervision. O.K.: Conceptualisation, Methodology, Writing - Review & Editing, Supervision. N.W.: Conceptualisation, Methodology, Writing - Review & Editing, Supervision.

## Competing interests

J.V.P. and N.W. are co-founders of genomiQa. O.K. has consulted for XING Technologies on development of diagnostic assays for HR deficiency. The remaining authors declare that there are no competing interests.
