## [Peer Review File · Nature Communications]

REVIEWER COMMENTS

Reviewer #1 (Remarks to the Author): Expert in computational genomics, bioinformatics, and cell type deconvolution

Bulk RNA-seq deconvolution using scRNA-seq reference for tumor samples has been an important field of research. Due to the heterogeneous gene expression of various cell types in the malignant cells and the tumor microenvironment, deconvolution has been a challenging problem. Various methods have been recently developed to address this issue, but the exact performance has not been extensively benchmarked. Tran et al. presented a study to benchmark the performance of these deconvolution methods.

Their framework of benchmarking was properly designed. They used a high-quality breast cancer scRNA-seq dataset to simulate pseudo-bulks. Notably, this dataset contains cells from normal epithelial lineages which resemble cancer epithelial cells and often cause problems to deconvolution. Additionally, deconvolution has been performed at a wide spectrum of granularity, which has rarely been done by previous studies.

However, there are several limitations in their design. Additional details of the benchmarking details also need some clarification.

Major comments:

1) The pseudo bulk and reference were generated using single cells generated by the same sequencing technology (10x Chromium) in a single study. Although heterogeneity in expression was captured by separating the dataset by patients, technical batch effects between scRNA-seq and bulk RNA-seq were under-represented. A good performance in this study, e.g. BayesPrism and DWLS, is necessary but not sufficient to generalize to real bulk data. Therefore, authors need to perform additional benchmarks for the best-performing tools in more realistic settings. I suggest additional benchmarks using TCGA breast cancer bulk RNA-seq by comparing the tumor purity estimates to those estimated by orthogonal experiments, such as CNV (ABSOLUTE), IHC, leukocyte methylation (LUMP) and/or marker expression (ESTIMATE). These estimates can be found from Aran, D. et al. Systematic pan-cancer analysis of tumour purity. Nat. Commun. 6:8971 doi: 10.1038/ncomms9971 (2015), and shall be done easily. Additional pseudo bulk generated from scRNA-seq using a different scRNA-seq assay in a different study is also recommended. One strategy to choose scRNA-seq dataset can be to select the one with the best overall correlation with the TCGA bulk RNA-seq.

2) How were the tumor cell types labeled? How does the granularity in the reference affect the prediction? Could additional clustering of tumor cells at higher granularity bring improvement to the deconvolution? Such study will give insights into how to best define tumor reference for deconvolution.

3) Will the use of marker genes increase the accuracy of BayesPrism and DWLS in distinguishing immune subsets? Feature selection would affect most of the deconvolution algorithm and is worth exploring in a benchmarking study.

Minor comments:

1) Some exact details of how pseudo bulk samples were generated need clarification. Were raw count or counts-per-10,000 normalized count (as mentioned in the data preprocessing section) used to perform summation?

2) I noticed that the author has made a few incorrect statements about bisqueRNA in the methods. First, the reference in the original BisqueRNA paper was not scTransformed, but rather count-per-million. scTransform generates residuals from the negative binomial regression, which were only used for clustering. Second, reference scRNA-seq should not be log-transformed, since $\log(A+B)$ does not equal $\log A + \log B$. Although I believe that this may not affect the overall results in this study, as Cobos et al. has shown Bisque performed similarly in log and linear space, I recommend the author to switch to non-log space across the board.

3) Why were 500 cells used to simulate the pseudo bulk? Does the total library size match a typical bulk data?

4) There are three levels of granularity of cell types annotated by the original study. How was cell type defined in the reference? My impression is that the author used the highest resolution, i.e. 49 subsets, and added up the imputed fractions to generate the fractions for cell type at major level. But this requires clarification.

5) MuSic allows modeling the information about the subject (patient ID) label. It is unclear how this information was used in the benchmark. Were patient IDs used for other methods?

4) It is worth double checking the benchmarking of CPM which predicts seemingly constant cell type fraction.

5) I recommend authors to show correlation coefficient and MSE on Extended data Fig. 8 directly. As data points with zeros values overlap severely.

6) A better visualization is needed for Figure 5d and e. One possibility is to compute the difference of accuracy between two methods for each cell type (y-axis) and then sort by cell type (x-axis)? Additionally, it is worth investigating the cause of error in the inference of cell type fractions. I suspect that the existence of a more similar cell type in the reference, the worse the performance, in which case

one may plot the error of each cell type of each method (y-axis) with respect to the correlation in gene expression to the most similar cell type (x-axis)?

7) SMOTE sampling is a linear extrapolation. The author should discuss this limitation, and mention other non-linear approaches that model the distribution of gene expression, such as variational autoencoders and generative adversarial networks.

8) There should be a mention about the method to generate tumor cell labels in the methods section, e.g., CNV score or somatic mutation.

9) The last sentence in the abstract appears to be a bit disconnected. It is unclear to me what “molecular features” refer to and which benchmark supports this conclusion.

Reviewer #2 (Remarks to the Author): Expert in bioinformatics, cancer genomics, single-cell RNA-seq, and tumour immune microenvironment

This manuscript presents comprehensive benchmarking of nine deconvolution methods that utilize scRNA-seq data as a reference to predict cell-type composition of the TME. The authors used pseudo-bulk RNA mixtures to perform multiple simulations that consider the effect of tumor purity, cancer subtypes and lineage levels of immune cells on deconvolution performance. They use original ideas in order to estimate the deconvolution performance, such as Aitchison distance and false positives/negatives predictions. The study points out to certain limitations while considering different deconvolution methods in the study of the TME. I think that the study focuses too much on the technical side and would benefit greatly from biological insights based on the new understanding it provides.

Major issues

- As noted, I think that the study focuses too much on the technicalities. The story of different epithelial cells subtypes is interesting but is discussed from a technical view. This could be analyzed from a cancer biology perspective and maybe provide insights to differences between BRCA subtypes.

- While I think the text is mostly straightforward, the figures are challenging and don't provide clear picture of the results. The figures are packed with too many panels and tables, and it is very hard to understand what the important message is.

- In figure 2a, the Aitchison distance used as a metric to compare deconvolution performance in different levels of tumor purity. Although its possible to evaluate the overall performance of each test unit by examining the median distance value of each box in the plot, the long whiskers suggest too much variance between the simulations. The concept of Aitchison distance is an interesting idea for comparing two sets of compositional data using a single value, yet it doesn't seem ideal for highlighting the effect of tumor purity on deconvolution results.

As you noted in other sections of the papers, the cancer cells display similar expression profiles to other non-cancer cell-types, such as normal epithelial cells. It seems that the large variation results from the proportion of epithelial cells in the simulated mixtures, and I can guess that there is a strong association between the Aitchison distance and the proportion of epithelial cells in the mixtures.

If this is true, the question of how tumor purity affects the overall deconvolution performance, is somewhat equivalent to asking how well a deconvolution method can distinguish between tumor cells to their closest normal cell-type.

- The main result in the paper is that deconvolution fails in tumors with high tumor purity. However, it doesn't seem that this is a strong result when looking at the figures. In 4/9 methods the median distance in 95% tumor purity is lower than in 5%. Moreover, it doesn't seem like there is a clear continuum also in the other methods. Consider CBX – the top and bottom purity levels are distinguished, but the rest seem pretty stable. If this statement is true only for the extremes, it doesn't seem that it can be generalized as presented in the abstract.

- Performance of deconvolution methods across normal epithelial lineages in different breast cancer molecular subtypes: This section includes some very interesting results and further analysis is required. It is important to understand why certain lineages of epithelial cells affect deconvolution performance in specific subtypes of breast cancer, whereas others do not. It is possible that certain lineage-specific signature genes overlap with certain markers of breast cancer subtypes. By answering this question, researchers might be able to choose the appropriate reference based on the type of cancer being studied.

Minor issues

- Page 16, Dataset selection and pre-processing: Elaborate on the data preprocessing. Did you use the raw data, or did you download a preprocessed version? Describe the preprocessing step (filtering, normalizations, cell-type annotation) chronologically. Describe the reasons for using certain filtering cutoff values and normalization methods.
- Figure 3: It might make sense to move this figure to the external data and using a different and more compact visualization to convey the key message.
- Figure 4: The concept of benchmarking deconvolution methods using false positives/negatives is an interesting idea. However the visualization in figure 4 is challenging to interpret. It might be possible to

use something like a ROC curve as inspiration for summarizing multiple confusion matrices with different prediction cutoffs for each cell-type.

- Data leakage between patients? Using SMOTE before splitting the data into training and testing might result in overfitting and optimistic performance of deconvolution methods. In case that a certain cell-type is absent in a sample. Did you synthetically “borrow” rare cell-types from one patient to another?

Reviewer #3 (Remarks to the Author): Expert in cancer genomics, tumour microenvironment, and cell type deconvolution

Tran et al. performed a comprehensive benchmark of cell type abundance deconvolution tools. This was done by applying and assessing several methods using breast cancer single cell RNA profiles from one study by Wu et al. The paper is well written, and the question they address is interesting. Deconvolution is a powerful approach to study cellular heterogeneity across large cohorts at a very limited cost, and benchmarks comparing deconvolution tools are important to know the limitations of the many methods published. However, this manuscript presents several shortcomings, which I have summarized below.

Major comments:

1. When performing a benchmark, one must ensure that the benchmarking methods have been implemented properly. For example, CIBERSORTx requires the creation of a signature matrix, which the authors of the present study have also used for EPIC. The authors of CIBERSORTx emphasize the importance of validating a signature matrix before applying it to other samples (Newman et al, 2019; Steen et al 2020). Which steps were taken here to make sure that the signature matrix is well-validated before applied to the test datasets? And in general, which steps were taken to ensure that each method was implemented and tested in a fair manner?
2. Limiting this study to (i) one cancer type, (ii) to reconstituted pseudo-bulks only, and (iii) to one breast cancer scRNA-seq dataset, makes this paper very limited, when it could in contrast be relevant to a wider scientific community where deconvolution is of interest. Since the analysis presented here is purely computational, it should be extended to additional BRCA scRNA-seq datasets available in the public domain (such as Bassez et al Nature Medicine 2021, and Pal et al EMBO 2021). More importantly, to be relevant to a broad journal such as Nature Communications, it should be extended to multiple cancer types where multitude of scRNA-seq datasets are available (including across cancer subtypes, to perform an analogous analysis as the one described here across BRCA subtypes). One such example would be non-small cell lung carcinoma (NSCLC), and there are others (melanoma, colorectal cancer, ...).

3. Finally, the gold-standard for benchmarking deconvolution methods is comparison to data obtained from an orthogonal method, such as flow cytometry or immunohistochemistry. Ideally the authors should have access to orthogonal data of samples profiled by both bulk and scRNA-seq, to properly assess the performance of these methods. This is a major point as this study is a purely benchmarking exercise with no new method or dataset generated, so it should be done thoroughly if to become a reference for users of these tools.

We thank the reviewers for their hopeful comments which have greatly improved our manuscript. Below we provide our response to each comment, with reviewers' comments in bold text.

REVIEWER #1

(Expert in computational genomics, bioinformatics, and cell type deconvolution)

Their framework of benchmarking was properly designed. They used a high-quality breast cancer scRNA-seq dataset to simulate pseudo-bulks. Notably, this dataset contains cells from normal epithelial lineages which resemble cancer epithelial cells and often cause problems to deconvolution. Additionally, deconvolution has been performed at a wide spectrum of granularity, which has rarely been done by previous studies. However, there are several limitations in their design. Additional details of the benchmarking details also need some clarification.

Response: We are glad the reviewer notes our framework as properly designed. We have answered questions about the limitations in the following comments.

Major comments:

1a) The pseudo bulk and reference were generated using single cells generated by the same sequencing technology (10x Chromium) in a single study. Although heterogeneity in expression was captured by separating the dataset by patients, technical batch effects between scRNA-seq and bulk RNA-seq were under-represented. A good performance in this study, e.g. BayesPrism and DWLS, is necessary but not sufficient to generalize to real bulk data. Therefore, authors need to perform additional benchmarks for the best-performing tools in more realistic settings. I suggest additional benchmarks using TCGA breast cancer bulk RNA-seq by comparing the tumor purity estimates to those estimated by orthogonal experiments, such as CNV (ABSOLUTE), IHC, leukocyte methylation (LUMP) and/or marker expression (ESTIMATE). These estimates can be found from Aran, D. et al. Systematic pan-cancer analysis of tumour purity. Nat. Commun. 6:8971 doi: 10.1038/ncomms9971 (2015) , and shall be done easily.

Response: We thank the reviewer for highlighting the need to benchmark the approaches in bulk RNA-seq data, and we appreciate the detailed suggestion of how to address this comment. Following the reviewer's suggestions, we have run all deconvolution methods on bulk RNA-seq data from TCGA breast cancer patients (n = 1,038) and assessed predictions of cancer cell proportions against tumour purity estimates produced by CNV (ABSOLUTE), H&E images, methylation (LUMP) and expression (ESTIMATE) from Aran et al. (**Rebuttal Figure 1**). This new analysis has enabled us to validate the deconvolution method's performance with real bulk mixtures.

Rebuttal Figure 1: Comparison of nine deconvolution approaches to four alternative methods to predict tumour purity in TCGA breast data. Scatter plot comparing purity levels determined by the nine deconvolution methods (columns) with the four alternative methods (ABSOLUTE, ESTIMATE, LUMP and Pathology) and their Consensus Purity Estimates (CPE) provided by Aran et al (rows). The data comprised $n = 968$ breast cancers from TCGA, with each sample coloured by the PAM50 subtype. The y-axis for each plot is the percentage of tumour cells predicted from the relevant deconvolution tool and the x-axis is the percentage of tumour cells predicted from the orthogonal method. Pearson's r correlation shown.

We found that Scaden, BayesPrism, MuSiC and DWLS achieved higher correlation with the four orthogonal tumour purity estimations. This result supports our findings using simulated bulk mixtures, as these deconvolution approaches had the lower RMSE scores compared to other methods. Similarly, CPM and Bisque showed worse correlation and higher RMSE scores. Previous studies have reported a positive correlation with tumour purity estimates from CNV (ABSOLUTE), H&E images, methylation (LUMP) and expression (ESTIMATE)^{1,2}. In agreement with this we found similar patterns with the four orthogonal tools when compared to each deconvolution approach (Rebuttal Figure 1). Therefore, we decided to limit the analysis in the paper to the results from ABSOLUTE.

As an additional benchmark using TCGA bulk RNA-seq data, we validated each method to predict lymphocyte proportions (T- and B-cell populations) against spatial tumour infiltrating lymphocyte (TIL) patterns derived from H&E images. To do this, we downloaded TIL estimations of 944 breast cancer patient patients in TCGA from Saltz et al³ (DOI: 10.1016/j.celrep.2018.03.086), which was produced by a trained deep learning model using H&E images. We subsequently used this information to calculate spatial TIL proportions and correlate with deconvolution lymphocyte proportions. Of note, the Saltz et

al. model was trained using H&E images manually annotated by pathologists and was shown to have strong correlation with cellular compositions derived by CIBERSORT in pan-cancer context³. We found that BayesPrism, DWLS and MuSiC show the best correlation with TIL estimation from H&E images, which agrees with these tools having the lower RSME using simulated bulk data. However, generally all deconvolution methods showed low correlation to TIL estimation from H&E (new Figure 2c-d). The poor correlation between TIL estimation from RNA-seq and H&E could be due to intra-tumour heterogeneity, as unlike the comparison to ABSOLUTE or LUMP, different pieces of the tumour tissue would have been used for RNA-seq and H&E. This conclusion is supported by the tumour purity estimation (Rebuttal Figure 1), where H&E images also show lower correlation.

Changes to manuscript:

- We have amended Figure 2 to include two new panels summarizing the bulk RNA-seq comparisons. Figure 2c is the comparison of the nine deconvolution approaches to ABSOLUTE for the estimation of tumour purity estimation, and Figure 2d is the comparison of the nine deconvolution approaches to the Saltz et al. method for the estimation of TILs from H&Es.
- We have amended the Results to include the TCGA bulk RNA-seq analysis (p.9): *"To confirm the deconvolution performance for major cancer and immune cell types using non-simulated real bulk RNA-seq data, we used data derived from 1,038 breast cancer patients from The Cancer Genome Atlas (TCGA) study^{6,39}. We compared predicted cancer cell proportions against tumour purity estimates produced by ABSOLUTE⁴⁰ in Aran et al⁴¹, which uses copy number variations, and predicted lymphocyte proportions (T-cells and B-cells) against tumour-infiltrating lymphocytes (TIL) estimates produced from H&E images using deep learning⁴². Consistent with the simulated bulk results, BayesPrism, Scaden and MuSiC showed the strongest performance for both cancer and lymphocyte predictions, evidenced by the highest Pearson's correlation coefficients and lowest RMSE scores (Fig. 2c-d). Similar to simulated bulk results, CBX, DWLS and hspe achieved better performance than Bisque, EPIC and CPM in predicting cancer proportions (Fig. 2c), however, except for hspe, all 5 methods over-predicted lymphocyte proportions (Fig. 2d). Generally, we did not observe patterns in prediction correlations across the PAM50 subtypes"*.
- We have amended the Discussion to include the TCGA bulk RNA-seq analysis. (p.16): *"The true test for deconvolution methods is how they perform on real bulk mixtures. Our validation analysis on 1,038 breast cancer samples from TCGA revealed that the three methods showing highest level of resilience against tumour purity (BayesPrism, Scaden and MuSiC) were also the top-performing methods in predicting tumour purity (compared to ABSOLUTE⁴¹ estimates) and tumour-infiltrating lymphocyte (TIL) content (compared to Saltz et al⁴² estimates). CBX and hspe also showed comparable performance to BayesPrism, Scaden, and MuSiC in predicting tumour purity. However, we note that most methods achieved sub-optimal performance with TIL estimation, which could have been a result of the different tumour regions used for RNA extraction and H&E imaging."*

- To address these comments, we have made substantial changes to the Methods section with the addition of new sections **Validation on real bulk mixtures from TCGA breast cancer samples** (p.26-27).

1b) Additional pseudo bulk generated from scRNA-seq using a different scRNA-seq assay in a different study is also recommended. One strategy to choose scRNA-seq dataset can be to select the one with the best overall correlation with the TCGA bulk RNA-seq.

Response: We agree with the reviewer that adding pseudobulks generated from different scRNA-seq datasets will help improve the generalizability of our study. This is similar to major comment 2 by Reviewer 3 who suggested we include two additional scRNA-seq datasets (Bassez et al and Pal et al). For detailed description of how we incorporated these two datasets please refer to the major comment 2 from Reviewer 3.

2) How were the tumor cell types labeled? How does the granularity in the reference affect the prediction? Could additional clustering of tumor cells at higher granularity bring improvement to the deconvolution? Such study will give insights into how to best define tumor reference for deconvolution.

Response: For the first question, epithelial cells were previously annotated using differential expression analysis on selected marker genes. These marker genes, as well as their average log-fold change, are provided In Supplementary Table 9 of Wu et al.⁴ (DOI: 10.1038/s41588-021-00911-1). Several cell-type-specific experts were also involved during the original annotation process. Additionally, in Wu et al. cancer cells were distinguished from normal epithelial cells based on Copy Number Variation (CNV) signal using inferCNV⁵, and cancer epithelial cells were subsequently clustered into 4 minor cell types using a unique set of differentially expressed genes for each cell type. This information is also provided in Supplementary Table 9 of Wu et al. We have revised the methods to clarify that we used the cell annotations provided by Wu et al⁶, Bassez et al⁷ and Pal et al⁸ without any alteration. The relevant section in Methods now reads (p.19): *“In this study, we collected raw gene counts and annotated single cell labels, including cancer cell labels which were inferred using Copy Number Variations (CNV), from the breast cancer scRNA-seq datasets in Wu et al¹⁵, Bassez et al³⁷, and Pal et al³⁸”*.

For the second question, we agree that deconvolving with more granular subtypes of cancer cells is possible. However, we believe this is more likely to lead to poorer performance as we will run into more rare cancer subtypes, with low cell counts that are harder to estimate with a fixed sample size. Additionally, our validation work on Pal et al⁸ showed less pronounced misprediction errors between cancer and normal epithelial cells. This was possibly due to cross-cohort heterogeneity and/or different approaches to cell annotation. Moreover, we would not be able to validate this hypothesis with BayesPrism and EPIC, as both methods specifically deconvolve cancer cells separately from other cell types and are currently not offering options to segment cancer populations further.

3) Will the use of marker genes increase the accuracy of BayesPrism and DWLS in distinguishing immune subsets? Feature selection would affect most of the deconvolution algorithm and is worth exploring in a benchmarking study.

Response: We agree with the reviewer that feature selection could influence deconvolution and have investigated whether the use of marker genes could improve deconvolution performance. In addition to BayesPrism, the methods MuSiC, Bisque, and hspe can also use marker genes as input. Therefore, we ran BayesPrism, MuSiC and hspe methods with cell-type-specific marker genes provided in Supplementary Table 9 of Wu et al. at all major, minor, and subset lineage levels. In addition to marker genes, BayesPrism and MuSiC also support the input of cell subtypes to help improve deconvolution performance. Thus we also ran these two methods with cell types where relevant.

Additionally, we ran DWLS using two signature matrices. DWLS uses either the R package Seurat or MAST to build its own internal signature matrix. The results we originally reported for DWLS were produced using Seurat, therefore we also re-ran DWLS using MAST.

Our results showed that including marker genes or cell subtypes does not significantly improve performance for BayesPrism, MuSiC and Bisque (Rebuttal Figure 2-4). For hspe, using default settings produced better deconvolution results than using marker genes (Rebuttal Figure 5). For DWLS, using Seurat for differential expression analysis seem to produce slightly worse results than using MAST (Rebuttal Figure 6). However, we note that it is difficult to compare predictions-vs-groundtruth scatterplots of DWLS, as the method produced a lot of very small predictions, e.g 0.00000001%. When comparing overall performance using Bray-Curtis dissimilarity and cell-type-specific performance using Root Mean Square Error (Rebuttal Figure 7), we found that while median Bray-Curtis scores of MAST-DWLS were lower than Seurat-DWLS, the variance was significantly slower for Seurat-DWLS. Additionally, cell-type-specific RMSE of Seurat-DWLS was significantly lower than MAST-DWLS. Due to these reasons, we have therefore decided to retain the original results generated from non-optimized execution of all methods.

Rebuttal Figure 2: Performance optimization results for BayesPrism. Scatter plots of predicted (y-axis) versus actual (x-axis) cell compositions of BayesPrism run with different settings (columns) and at different lineage levels (rows). BayesPrism was executed using either all default settings (left column), with cell states (i.e. cell subtypes, middle column), or with both cell states and marker genes (right column). Each point represents a test mixture component, with its colour representing one of eight test patients (from which single cells were used to generate the mixture). Dotted 45-degree diagonal line represents perfect prediction where predicted proportions match actual proportions. All cell types are represented in each plot, and each scatterplot is annotated with Pearson's correlation coefficient (r) and Root Mean Square Error ($rmse$).

Rebuttal Figure. 3: Performance optimization results for MuSiC. Scatter plots of predicted (y-axis) versus actual (x-axis) cell compositions of **MuSiC** run with different settings (columns) and at different lineage levels (rows). **MuSiC** was executed using either all default settings (left column) or with marker genes (right column). Each point represents a test mixture component, with its colour representing one of eight test patients (from which single cells were used to generate the mixture). Dotted 45-degree diagonal line represents perfect prediction where predicted proportions match actual proportions. All cell types are represented in each plot, and each scatterplot is annotated with Pearson's correlation coefficient (r) and Root Mean Square Error ($rmse$).

Rebuttal Figure 4: Performance optimization results for Bisque. Scatter plots of predicted (y-axis) versus actual (x-axis) cell compositions of **Bisque** run with different settings (columns) and at different lineage levels (rows). **Bisque** was executed using either linear gene counts (left column), using logged gene counts (middle column), or using linear counts and with marker genes (right column). Each point represents a test mixture component, with its colour representing one of eight test patients (from which single cells were used to generate the mixture). Dotted 45-degree diagonal line represents perfect prediction where predicted proportions match actual proportions. All cell types are represented in each plot, and each scatterplot is annotated with Pearson's correlation coefficient (r) and Root Mean Square Error (rmse).

Rebuttal Figure. 5: Performance optimization results for hspe. Scatter plots of predicted (y-axis) versus actual (x-axis) cell compositions of hspe run with different settings (columns) and at different lineage levels (rows). hspe was executed using either all default settings (left column) or with marker genes (right column). Each point represents a test mixture component, with its colour representing one of eight test patients (from which single cells were used to generate the mixture). Dotted 45-degree diagonal line represents perfect prediction where predicted proportions match actual proportions. All cell types are represented in each plot, and each scatterplot is annotated with Pearson's correlation coefficient (r) and Root Mean Square Error (rmse).

Reviewers-only-Figure 6

Rebuttal Figure 6: Performance optimization results for DWLS. Scatter plots of predicted (y-axis) versus actual (x-axis) cell compositions of DWLS run with different settings (columns) and at different lineage levels (rows). DWLS was executed using either Seurat (left column) or with MAST (right column) for differential expression analysis. Each point represents a test mixture component, with its colour representing one of eight test patients (from which single cells were used to generate the mixture). Dotted 45-degree diagonal line represents perfect prediction where predicted proportions match actual proportions. All cell types are represented in each plot, and each scatterplot is annotated with Pearson's correlation coefficient (r) and Root Mean Square Error (rmse).

Rebuttal Figure. 7: Deconvolution performance DWLS using either Seurat or MAST for differential expression analysis. a) Bray-Curtis dissimilarity predicted and ground truth cell compositions across 7 tumours purity levels (from 5% to 95%, 15% interval) with 2,000 artificial bulk mixtures per purity level. Upper and lower whiskers depict Bray-Curtis values outside of the centre 50%. **b)** Root mean square errors (RMSE) between predicted and actual cell compositions, aggregated by cell type. Seven tumour purity levels are shown (from 5% to 95%, 15% interval). Darker shade of red represents higher RMSE values, with numeric RMSE values shown. For both **(a)** and **(b)**, deconvolution methods in display are DWLS version that uses Seurat for differential expression analysis (Seurat-DWLS, top panel) or DWLS version that uses MAST for differential expression analysis (MAST-DWLS).

Minor comments:

1) Some exact details of how pseudo bulk samples were generated need clarification. Were raw count or counts-per-10,000 normalized count (as mentioned in the data preprocessing section) used to perform summation?

Response: We thank the review for this suggestion and have added further details to the methods section of pseudobulk generation as follows:

(p.19-20): “For all 3 datasets, we normalized downloaded raw gene counts by counts-per-10,000 using Seurat v3 NormalizeData() function with normalisation method set to relative counts (RC), which applied counts-per-10,000 normalisation without log transformation. The resulting normalized counts were then used for downstream benchmarking analyses.”

2) I noticed that the author has made a few incorrect statements about bisqueRNA in the methods. First, the reference in the original BisqueRNA paper was not scTransformed, but rather count-per-million. scTransform generates residuals from the negative binomial regression, which were only used for clustering. Second, reference scRNA-seq should not be log-transformed, since $\log(A+B)$ does not equal $\log A + \log B$. Although I believe that this may not affect the overall results in this study, as Cobos et al. has shown Bisque performed similarly in log and linear space, I recommend the author to switch to non-log space across the board.

Response: We thank the reviewer for picking this up. We have re-run Bisque with non-logged counts across all experiments and compared this to our original analysis with logged counts (**Rebuttal Figure 8**). As suspected by the reviewer, results produced by the new non-logged version of BisqueRNA has not altered the overall results, however we have replaced the data in the manuscript with non-logged counts.

Rebuttal Figure. 8: Cell-type-specific performance optimization results for Bisque RMSE between predicted and actual cell compositions, aggregated by cell type. Seven tumour purity levels are shown (from 5% to 95%, 15% interval). Left heatmap represents results obtained when Bisque was run using linear gene counts, and right heatmap represents results obtained when Bisque was run using logged gene counts. Shade of red represents higher RMSE values, with numeric RMSE values shown. CAFs: Cancer Associated Fibroblasts, PVL: Perivascular-like, RMSE: Root Mean Square Error.

Changes to manuscript:

We have included the new Bisque analysis using non-logged counts and have amended:

- The methods to detail the use of non-logged counts in both bulk mixtures and single-cell reference profiles. Methods Section (p.29) *“we chose not to process single-cell data for Bisque using the SCTransform function, but instead kept gene counts in linear counts, similar to other methods, and only applied min-max scaling as recommended by Bisque’s instructions.”*
- All figures with the new Bisque analysis using non-logged counts have been revised: Figures 1-5, and Extended Data Figures 1-7.

3) Why were 500 cells used to simulate the pseudo bulk? Does the total library size match a typical bulk data?

Response:

The recommended total RNA input for bulk RNA-seq library preparation can be as low as 10ng (SMART-seq) or 25ng (Illumina Stranded mRNA prep), which translates to 500 or 1,250 cells given the assumption that a typical mammalian cell contains 10-30 pg of total RNA. We chose to use 500 cells per pseudobulk to ensure we had enough cells for the simulated pseudobulk mixtures, particularly for the cancer purity experiments (where cancer cells made up to 95% of the mixture). As some samples contained between 500-1,000 cells for the most prevalent cell type that was used as the cell number for SMOTE oversampling, we were unable to SMOTE more cells than this number. We wanted to avoid sampling some cells twice, which would have been required if we used a higher number of cells per pseudo-bulk.

In terms of the library size, the average number of reads per cell is around 7,000 in Wu et al⁶. This adds up to around 3.5 million reads per pseudobulk, which is less than 25-50 million reads for a typical pseudobulk. We did, however, normalise scRNA-seq read counts to counts-per-10,000, which was comparable to using transcripts-per million normalised read counts in real bulk. We note that the smaller total library size means genes with lower expression would be less represented compared to bulk RNA-seq, which is a potential limitation of our study design. However, due to cell number constraints outlined above, we are unable to adjust the design.

Changes to manuscript:

- We have included text in the Discussion section to mention small number of cells per pseudobulk and library size as potential limitations (p.18): *“Lastly, in this study we fixed cell count of all pseudobulk mixtures at 500. This was mainly to ensure that we did not resample the same cell twice for a mixture, as some samples only contained between 500 and 1,000 for the most prevalent cell type. With the average number of reads per cell of approximately 7,000 in Wu et al¹⁵, a 500-cell mixture would have around 3.5 million reads per pseudobulk, which is less than the typical amount of 25-50 million reads. This small library size could have led underrepresentation of low expression compared to real bulk data.”*
- We have also added our description of library size of pseudobulk in **Supplementary Information**, under **Supplementary Note 2/Library size of simulated bulk mixtures**.

4) There are three levels of granularity of cell types annotated by the original study. How was cell type defined in the reference? My impression is that the author used the highest resolution, i.e. 49 subsets, and added up the imputed fractions to generate the fractions for cell type at major level. But this requires clarification.

Response: We did not use highest-resolution cell labels and add up fractions to higher lineage level. Instead, we conducted different experiments with different single-cell reference matrices using the corresponding cell types. For example, we only used B cells, T cells, and Myeloid at the major immune lineage level.

Changes to manuscript:

We have added additional clarifications to each experiment in the Methods section, including **Variable tumour-purity mixtures** (p.24), **Normal epithelial lineages mixtures** (p.24), and **Immune lineages mixtures** (p.24-25).

5) MuSiC allows modeling the information about the subject (patient ID) label. It is unclear how this information was used in the benchmark. Were patient IDs used for other methods?

Response: We did indeed use patient ID for MuSiC across all experiments. This input information is required for MuSiC, as the method relies on it to calculate inter-patient variance of gene expressions during the deconvolution process. Apart from MuSiC, no other methods require patient ID.

Changes to manuscript:

We have added additional text in the Methods section to clarify this (p.31): *“Notably, MuSiC implements a multi-subject gene-weighting approach, which dynamically penalises genes with low between-subject variance for the single-cell reference profiles. This feature is a core step of MuSiC deconvolution process, and we therefore chose to utilise it by including matching patient IDs for the single-cell reference.”*

6) It is worth double checking the benchmarking of CPM which predicts seemingly constant cell type fraction.

Response: We agree this result was surprising. We have subsequently had another bioinformatician independently run CPM and they achieved the same results.

7) I recommend authors to show correlation coefficient and MSE on Extended data Fig. 8 directly. As data points with zeros values overlap severely.

Response: We agree with and thank the reviewer for this helpful suggestion.

Changes to manuscript: We have added correlation coefficient and RMSE statistics to all scatter plot figures, i.e. Figures 2c-d and Extended Data Figure 8.

8) A better visualization is needed for Figure 5d and e. One possibility is to compute the difference of accuracy between two methods for each cell type (y-axis) and then sort by cell type (x-axis)?

Additionally, it is worth investigating the cause of error in the inference of cell type fractions. I suspect that the existence of a more similar cell type in the reference, the worse the performance, in which case one may plot the error of each cell type of each method (y-axis) with respect to the correlation in gene expression to the most similar cell type (x-axis)?

Response: We thank the reviewer for this suggestion. However, we believe the lineage depiction in Figure 5d and 5e are crucial for understanding cell-type-specific deconvolution performance across lineage levels. This information is also complemented by the descriptive statistics in Supplementary Table 3.

9) SMOTE sampling is a linear extrapolation. The author should discuss this limitation, and mention other non-linear approaches that model the distribution of gene expression, such as variational autoencoders and generative adversarial networks.

Response: We thank the reviewer for pointing this out and agree with the reviewer that non-linear generative approaches such as variational autoencoders and generative adversarial networks could better model the underlying distribution of single-cell gene expressions compared to SMOTE. We used SMOTE in this paper because its linear method to synthesize new data aligns with the linear scale of gene counts. However, we recognise that there are other non-linear methods and comparisons between them and SMOTE are crucial. We also acknowledge that there have been several variations of SMOTE capable of applying conditional modelling on the synthesis process. We have followed up on the reviewer's suggestion and added a discussion point on potential limitations of SMOTE and recommendations for future studies to leverage generative methods for single-cell data synthesis.

Changes to manuscript:

We have added the following text to the Discussion section (p.15): *“On this note, we acknowledge that a potential limitation of the SMOTE version we used is it does not attempt to model the underlying distribution of single-cell gene distributions. On the other hand, non-linear and generative deep learning methods such as variational autoencoder⁴⁴ and generative adversarial network⁴⁵ can learn the distribution of each class and enforce synthesized samples to fit in such distributions. A notable example is DeepSMOTE⁴⁶, which combines the Cartesian distance algorithm of SMOTE with a variational autoencoder architecture. We recommend future studies explore the potentials of generative deep learning methods in single-cell data synthesis.”*

10) There should be a mention about the method to generate tumor cell labels in the methods section, e.g., CNV score or somatic mutation.

Response: We did not assign cell labels for either tumour cells or any other cell types in this study. For all scRNA-seq datasets (Wu et al⁶, Bassez et al⁷, and Pal et al⁸), we used the cell annotations provided by the related studies, which include tumour cell labels. We do note that in all of these publications, cancer

epithelial cells were separated from normal cells by the method inferCNV using Copy Number Variation (CNV) abundance.

Changes to manuscript:

We have added clarification in our Methods section to emphasize that we used the original annotations from each study.

(p.19): *“In this study, we collected raw gene counts and annotated single cell labels, including cancer cell labels which were inferred using Copy Number Variations (CNV), from the breast cancer scRNA-seq datasets in Wu et al¹⁵, Bassez et al³⁷, and Pal et al³⁸.”*

11) The last sentence in the abstract appears to be a bit disconnected. It is unclear to me what “molecular features” refer to and which benchmark supports this conclusion.

Response: We agree with the reviewer and have revised this sentence to refer to *“tumour cell compositions”* as opposed to *“sample molecular features”*.

REVIEWER #2

(Expert in bioinformatics, cancer genomics, single-cell RNA-seq, and tumour immune microenvironment)

Major issues

1. As noted, I think that the study focuses too much on the technicalities. The story of different epithelial cells subtypes is interesting but is discussed from a technical view. This could be analyzed from a cancer biology perspective and maybe provide insights to differences between BRCA subtypes.

from a cancer biology perspective and maybe provide insights to differences between BRCA subtypes.

Response: We agree with the reviewer that there is a biological phenomenon underlying the interaction between breast cancer's PAM50 subtypes and misprediction of different epithelial cell subtypes (Luminal Progenitor, Mature Luminal, and Myoepithelial). We have added a section in our discussion commenting on the transcriptomic similarities between cancer subtypes and normal epithelial cell types. We discussed that these similarities support hypotheses about cell-of-origin for the different cancer subtypes. In addition, to further highlight biological differences that may exist, in the new Figure panels 2c and d, we have colored the samples by PAM50 status (Basal, Her2, LumA, LumB and Normal).

Changes to manuscript:

We have added the following text to the Discussion (p.17): *“We also found that cancer cells were mis-predicted as different subtypes of normal epithelial cells depending on the breast cancer subtype, observed in the predictions BayesPrism, Scaden, MuSiC, CBX, DWLS, hspe and EPIC. This observation supports that transcriptional similarities between normal epithelial cell subtypes and different cancer subtypes, which could be reflective of the cancer cell of origin. Luminal progenitor cells were generally overpredicted as TNBC cancer samples, aligning with the hypothesis that basal-like cancers, which overlap with TNBC subtyp⁵³, originate from luminal progenitor cells^{54,55}. Mature luminal cells were generally overpredicted for ER+ cancer samples. These cells are hormone receptor positive and are reported to have molecular profiles closest to luminal A cancers⁵⁴, which are generally ER+/PR+⁵³, although the cell of origin for this subtype is yet to be definitively confirmed⁵⁵. Finally, HER2+ cancers had a less definitive normal cell type mis-prediction, which varied depending on the deconvolution method, although mostly reflected luminal cell lineage rather than myoepithelial lineage. This aligned with the hypothesis that HER2+ cancers arise through HER2 (ERBB2) amplification in cells committed to the luminal lineage^{54,55}.”*

2. While I think the text is mostly straightforward, the figures are challenging and don't provide clear picture of the results. The figures are packed with too many panels and tables, and it is very hard to understand what the important message is.

Response: We thank the reviewer for the constructive comment to improve our figures. To simplify the figures, we have revised Figures 3 and 4 as the reviewer suggested (see our response to minor comments 2 and 3).

3. In figure 2a, the Aitchison distance used as a metric to compare deconvolution performance in different levels of tumor purity. Although its possible to evaluate the overall performance of each test unit by examining the median distance value of each box in the plot, the long whiskers suggest too much variance between the simulations.

The concept of Aitchison distance is an interesting idea for comparing two sets of compositional data using a single value, yet it doesn't seem ideal for highlighting the effect of tumor purity on deconvolution results.

Response: We thank the reviewer for highlighting the high variance of Aitchison distance distributions in Fig. 2b. After closer inspection we discovered that the variance was due to the presence of '0' values, since higher tumour purity naturally introduces a higher number of 0% across non-cancer cell types. When we used Aitchison distance, we first apply centered-log-ratio transformation on cell-type proportions, which requires 0% proportions to be replaced by a small number of to avoid zero-division in log ratios. We originally replaced 0% by 0.001% (i.e. 0.0001), which caused zeros to the transformed to large negative numbers in log-ratio scale. We have partially addressed the high variance issue by replacing 0% by 0.01% (i.e. 0.001) instead of 0.0001%, which reduced the magnitude of large negative number resulting from log-ratio transformation. Yet, for the tumour purity experiments where mixtures with high tumour purity have small contributions of all other cell types, comparing performance of methods using Aitchison distance remained problematic. Therefore, to revise our assessment of the impact of tumour purity on cell type deconvolution we have replaced Aitchison distance with another compositional metric, Bray-Curtis dissimilarity, which does not require log transformation.

We do, however, continue to use Aitchison distance to compare methods' performance across immune lineage levels, as the number of components (i.e. cell types) change between lineage levels. This is because Aitchison distance has been shown to have robust sub-compositional dominance⁹ (i.e. performance on all components and performance on a subset of components are comparable), which makes it a reasonable metric to compare deconvolution performance on compositions with different components. Additionally, in this analysis we fixed tumour purity at 50%, which prevented the zero-inflation issue caused by higher tumour purity discussed above.

Changes to manuscript:

- We have replaced Aitchison distance with Bray-Curtis dissimilarity for the assessment of the impact of tumour purity in Figure 2a and Extended Data 1a.
- We have used Bray-Curtis dissimilarity in the new analysis of additional data sets in Extended Data 2a and 3a.
- We have added a paragraph to **Methods/Evaluation metrics** (p.31-32) to describe Bray-Curtis dissimilarity.

4. As you noted in other sections of the papers, the cancer cells display similar expression profiles to other non-cancer cell-types, such as normal epithelial cells. It seems that the large variation results from the proportion of epithelial cells in the simulated mixtures, and I can guess that there is a strong association between the Aitchison distance and the proportion of epithelial cells in the mixtures.

If this is true, the question of how tumor purity affects the overall deconvolution performance, is somewhat equivalent to asking how well a deconvolution method can distinguish between tumor cells to their closest normal cell-type.

Response: We thank the reviewer for this keen observation. For BayesPrism, Scaden, MuSiC, CBX, DWLS, hspe and EPIC, the ability to distinguish between tumour cells and its closest minor cell type of normal epithelial cells (including luminal progenitors, mature luminal, and myoepithelial) was the key driver behind poor performance at high tumour purity levels. We have revised the relevant Discussion section to highlight this phenomenon more clearly.

Changes to manuscript:

- We have updated the relevant Discussion section as described in our response to major comment 1 above.

5. The main result in the paper is that deconvolution fails in tumors with high tumor purity. However, it doesn't seem that this is a strong result when looking at the figures. In 4/9 methods the median distance in 95% tumor purity is lower than in 5%. Moreover, it doesn't seem like there is a clear continuum also in the other methods. Consider CBX – the top and bottom purity levels are distinguished, but the rest seem pretty stable. If this statement is true only for the extremes, it doesn't seem that it can be generalised as presented in the abstract.

Response: We agree with the reviewer that using the Aitchison distance values (as per this reviewer's question 3) shown in the previous version of our manuscript, overall performance of CBX (and also Bisque) were quite indistinguishable across most tumour purity levels. We have followed up on the reviewer's comment and replaced Aitchison distance with Bray-Curtis dissimilarity to assess the impact of tumour purity on deconvolution.

We believe the use of Bray-Curtis dissimilarity has been able to highlight the difference in overall deconvolution performance across tumour purity levels. More specifically, results from Wu et al (Figure 2a) show that BayesPrism, MuSiC and Scaden were the top-performing methods, with BayesPrism and MuSiC performing slightly better with higher tumour purity and Scaden performing better with the top and bottom purity levels. hspe showed the same pattern as BayesPrism and MuSiC, however its Bray-Curtis dissimilarity was not as low as either BayesPrism, Scaden, or MuSiC across all tumour purity levels. By contrast, performance of CBX, DWLS, EPIC, CPM decreases with tumour purity levels. Furthermore, by validating our findings on Bassez et al and Pal et al, we were also able to generalise the robustness of BayesPrism, Scaden and MuSiC against variable tumour purity levels across all 3 datasets.

Additionally, we also explored how well methods predicted cancer populations across tumour purity levels (Extended Data Fig. 5) and found that performance of even the best performing methods (BayesPrism, Scaden and MuSiC) still dropped at the extreme ends of tumour purity (5% and 95%). This is phenomenon is most apparent for BayesPrism at 95% and MuSiC at 5%.

We have updated the Results and Discussion as described in our response to the previous comment on Aitchison distance to reflect these patterns. The results in Extended Data Fig. 5 have also been added to

the manuscripts at p.8: *“We investigated this further by examining the distributions of predicted cancer proportions, aggregated by tumour purity levels (Extended Data Fig. 5). The results showed that most of Bisque predictions centred around 20% tumour purity in Wu et al. and Bassez et al. datasets, and dramatically shifted 65% in the Pal et al dataset. By contrast, predicted cancer proportions of the top performing methods (BayesPrism, Scaden, and MuSiC) generally followed their corresponding tumour purity levels, highlighting the robustness of these methods across datasets.”*

6. Performance of deconvolution methods across normal epithelial lineages in different breast cancer molecular subtypes: This section includes some very interesting results and further analysis is required. It is important to understand why certain lineages of epithelial cells affect deconvolution performance in specific subtypes of breast cancer, whereas others do not. It is possible that certain lineage-specific signature genes overlap with certain markers of breast cancer subtypes. By answering this question, researchers might be able to choose the appropriate reference based on the type of cancer being studied.

Response: We agree with the reviewer that different lineages of normal epithelial cells being mispredicted as cancer cells dependent on molecular subtype could be a result of transcriptional similarities between cancer cells and the normal cell lineage they originated from. For example, Lim et al¹⁰ and Fu et al¹¹ suggested that basal-like cancer cells originate from luminal progenitor cells in triple-negative breast cancer (TNBC), which lends strength to our findings showing luminal progenitor overpredicted at almost the same magnitude that cancer was underpredicted for TNBC samples. We have added a detailed discussion on this biological interaction in the Discussion as described in our response to the previous comment on Aitchison distance.

Minor issues

1. Page 16, Dataset selection and pre-processing: Elaborate on the data preprocessing. Did you use the raw data, or did you download a preprocessed version? Describe the preprocessing step (filtering, normalizations, cell-type annotation) chronologically. Describe the reasons for using certain filtering cutoff values and normalization methods.

Response: We collected raw gene counts from each study and applied counts-per-10,000 normalisation before any downstream analysis.

Changes to manuscript: We have revised the Methods section to provide clarification to all points suggested by the reviewer. (p.19): *“In this study, we collected raw gene counts and annotated single cell labels, including cancer cell labels which were inferred using Copy Number Variations (CNV), from the breast cancer scRNA-seq datasets in Wu et al¹⁵, Bassez et al³⁷, and Pal et al³⁸.”*

2. Figure 3: It might make sense to move this figure to the external data and using a different and more compact visualization to convey the key message.

Response: We appreciate this keen observation from the reviewer and agree that Figure 3 could be simplified.

Changes to manuscript:

We have revised Figure 3a to summarise the performance of each method for each cell type. We have reduced the number of boxplots to only include cancer epithelial and the 3 minor cell types of normal epithelial (now Figure 3b).

3. Figure 4: The concept of benchmarking deconvolution methods using false positives/negatives is an interesting idea. However the visualization in figure 4 is challenging to interpret. It might be possible to use something like a ROC curve as inspiration for summarizing multiple confusion matrices with different prediction cutoffs for each cell-type.

Response: We thank the review for this helpful feedback. It is true that ROC curves can be drawn for any score that is monotonically related to our predictions (present/absent). Unfortunately, in our case, we cannot draw ROC curves as, for example, a 10% threshold does not mean that we classify everything below 10% as absent and vice versa. Rather, the 0.1%, 1%, and 10% are severity levels of false positive (predicting a cell type is present while it's absent) and false negative predictions (predicting a cell type is absent while it is present). We have, however, substantially altered Figure 4 to better convey the results.

Changes to manuscript:

- We have included confusion matrix in Figure 4a along with false positive and false negative rates statistics across all cell types for each method. We have summarised the false positive and false negative rates for each method in Figure 4b and c.
- We have moved the cell-type-specific illustrations of false positive and false positive rates to Extended Data Figure 7.

Additionally, we have also explored plotting False Positive Rates (FPR) against True Positive Rates (TPR, equal to $1 - \text{False Negative Rates}$) at different severity levels of FPR and TPR and presented the visualization in Rebuttal Figure. 9, compared to which we believe the revised Figure 4 and Rebuttal Figure 7 present the same information more clearly and succinctly.

Rebuttal Figure 9. True positive rates (TPR) against False Positive rates (FPR) at different cumulative severity levels. Scatter plot of FPR (x-axis) versus TPR (y-axis, equal to $1 - \text{False Negative Rates}$) across 2,000 pseudobulk mixtures with 50% tumour purity level and nine major cell types. In (a), FPR were calculated at different cumulative levels of predictions: 0.1%-1% (false positive with predictions between 0.1% and 1%), 0.1%-10% (false positive with predictions between 0.1% and 10%) and 0.1%-100% (false positives with predictions between 0.1% and 100%). In (b), TPR ($1 - \text{FNR}$) were calculated at different cumulative levels of groundtruth: 0.1%-1% (false negative with groundtruth between 0.1% and 1%), 0.1%-10% (false negative with groundtruth between 0.1% and 10%) and 0.1%-100% (false negative with groundtruth between 0.1% and 100%). These levels are represented by either circle (0.1%-1%), diamond (0.1%-10%), or square (0.1%-100%). Methods are represented by colours.

4. Data leakage between patients? Using SMOTE before splitting the data into training and testing might results in overfitting and optimistic performance of deconvolution methods. In case that a certain cell-type is absence in a sample. Did you synthetically “borrow” rare cell-types from one patient to another?

Response: We did not borrow rare cell types between patients during the SMOTE process. If a cell type is not present for a patient, we simply did not “SMOTE” it for such patient, resulting in a cell-type count of 0 after SMOTE. We did, however, ensure that all cell types are present in both training and testing data by discarding cell types that only exist in either training or testing data.

Changes to manuscript:

We have revised the Methods to clarify this. (p.22s): “To preserve inter-patient variability, we conducted SMOTE separately for each patient, i.e. synthesised cells are created for each patient using only the patient’s original cells. This means if a cell type is not present for a patient, its cell count after SMOTE remains 0.”

REVIEWER #3

(Expert in cancer genomics, tumour microenvironment, and cell type deconvolution)

Major comments:

1a. When performing a benchmark, one must ensure that the benchmarking methods have been implemented properly. For example, CIBERSORTx requires the creation of a signature matrix, which the authors of the present study have also used for EPIC. The authors of CIBERSORTx emphasize the importance of validating a signature matrix before applying it to other samples (Newman et al, 2019; Steen et al 2020). Which steps were taken here to make sure that the signature matrix is well-validated before applied to the test datasets?

Response: We thank the reviewer for this great suggestion and have followed up validating signature matrices produced by single-cell-based deconvolution methods. This applies to signature matrices from all methods besides Scaden and EPIC in our study.

Overall, Newman et al and Steen et al suggested three avenues to validate single-cell-derived signature matrices: 1) check whether cell-type-specific marker genes are present, 2) evaluate each method's performance using its own signature matrix on pseudobulks made from nonlog linear counts, which is what our study presented, and 3) compared deconvolution performance against orthogonal methods such as flow cytometry or immunohistochemistry (IHC). As per point (2), reviewer 2 has accurately pointed out that we previously used logged counts for Bisque, which we have since corrected by re-running Bisque using nonlog counts (minor comment 2). As per point (3), we have now comprehensively validated the benchmarked deconvolution methods against tumour purity from copy number variations and spatial tumour infiltrating lymphocytes produced by deep learning model³. Please refer to major comment 1 from Reviewer 1 where we have described this new work.

1b. And in general, which steps were taken to ensure that each method was implemented and tested in a fair manner?

Response: We have treated deconvolution methods as fairly as we can by downloading the original code and executing each method by strictly following their execution instructions and using their default parameters as much as possible. On this note, we originally applied logarithm transformation on single-cell reference profiles of Bisque before deconvolution, which was incorrect and was pointed out by Reviewer 1 (minor comment 2). We have fixed this issue and replaced Bisque results with those generated using linear gene counts throughout the manuscript.

Additionally, we have followed up on major comment 1 from Reviewer 1 on whether deconvolution performance can be optimised further by the additional input of marker genes and cell subtypes where applicable. We found that except for hspe which performed better with default settings, adding marker genes and/or cell subtypes did not result in significant performance improvement. Please see major comment 3 from Reviewer 1 for more details.

Changes to manuscript:

We have updated the Methods section to include all deviations from default execution settings.

(p.29): *“we chose not to process single-cell data for Bisque using the SCTransform function, but instead kept gene counts in linear counts, similarly to other methods, and only applied min-max scaling as recommended by Bisque’s instructions”*

(p.30) *“CPM instructions recommend choosing a neighborhoodSize value representing the maximum number of cells of the same cell type surrounding any random cell in the cell-state space. As our cell-state space was a dense UMAP distribution, we aimed to select the highest possible value for neighborhoodSize. We followed CPM’s tutorial where neighborhoodSize is roughly one-fourth of the count of the smallest cell type. As our least abundant cell type was 1,330 (normal epithelial cells), we chose neighborhoodSize=250”*

2. Limiting this study to (i) one cancer type, (ii) to reconstituted pseudo-bulks only, and (iii) to one breast cancer scRNA-seq dataset, makes this paper very limited, when it could in contrast be relevant to a wider scientific community where deconvolution is of interest. Since the analysis presented here is purely computational, it should be extended to additional BRCA scRNA-seq datasets available in the public domain (such as Bassez et al Nature Medicine 2021, and Pal et al EMBO 2021). More importantly, to be relevant to a broad journal such as Nature Communications, it should be extended to multiple cancer types where multitude of scRNA-seq datasets are available (including across cancer subtypes, to perform an analogous analysis as the one described here across BRCA subtypes). One such example would be non-small cell lung carcinoma (NSCLC), and there are others (melanoma, colorectal cancer, ...).

Response: In regard to point (i), we feel that focusing on breast cancer only is the best approach for the paper. This is because, while addressing major comment 1 from Reviewer 2 we have included more biological insights that are specific to breast cancer. Additionally, while addressing point (ii) and (iii) of this comment we have made a number of substantial additions to the manuscript including increasing the number of Extended Data Figures. This means the paper is now quite large and the addition of other tumour types would reduce the granularity of analysis, and thus is out of scope for the current study. While we do appreciate the request for other tumour types, it is also challenging to know what types to include or we may be restricted to an eclectic assortment of tumours based on what data is available. We feel that focusing on breast cancer does not necessarily make this study less interesting to the broad readership, as plenty of activities start in the most studied cancers then permeate to other cancer types in subsequent investigations.

For point (ii) we have included an analysis of bulk RNA-seq from >1,000 breast cancer samples from TCGA. Please see major comment 1 for Reviewer 1 for more details.

For point (iii) we thank the reviewer for the suggestion of incorporating using the Bassez et al 2021 and Pal et al 2021 single cell datasets to verify our findings. We have replicated the investigation into the influence of tumour purity on deconvolution using these 2 datasets. Encouragingly, these results showed similar patterns as those from the original dataset (Wu et al) and have sustainably strengthened our findings.

Changes to manuscript:

- We have included new figure panels, Figure 2c and 2d showing the results from the analysis of bulk tissues from TCGA.
- We have included new Extended Data Figure 2 and -3 showing the results from Bassez et al 2021 and Pal et al 2021 single cell datasets.
- We have included a new section in the Results on p.7 “**Confirmation of the performance of TME deconvolution methods**”.
- We have included text in the Methods to describe the analysis of the Bassez et al 2021 and Pal et al 2021 single cell dataset.
- We have also updated the Discussion to reflect the inclusion of Bassez et al and Pal et al data. (p.16): *“Furthermore, our validation work on Bassez et al³⁷ and Pal et al³⁸ confirmed BayesPrism, MuSiC and Scaden are resilient against variable tumour purity, and their good performance on deconvolving major immune cell types are generalisable across datasets. In Pal et al³⁸, where normal epithelial cell labels were provided, we also confirmed the normal-cancer misprediction for MuSiC, CBX, DWLS, and hspe.”*
- We have amended the abstract to refer to the additional datasets *“a finding that was validated in two independent datasets”*

3. Finally, the gold-standard for benchmarking deconvolution methods is comparison to data obtained from an orthogonal method, such as flow cytometry or immunohistochemistry. Ideally the authors should have access to orthogonal data of samples profiled by both bulk and scRNA-seq, to properly assess the performance of these methods. This is a major point as this study is a purely benchmarking exercise with no new method or dataset generated, so it should be done thoroughly if to become a reference for users of these tools.

Response: We agreed with the reviewer that validation against orthogonal methods is crucial in deconvolution benchmarking. We have conducted validation of deconvolution methods against orthogonal methods (tumour purity estimations produced by ABSOLUTE in Aran et al² and tumour-infiltrating lymphocyte estimations produced by a deep learning model from Saltz et al³). This work is described in detail in our response to major comment 1 from Reviewer 1.

References

1. Yoshihara, K. *et al.* Inferring tumour purity and stromal and immune cell admixture from expression data. *Nat. Commun.* **4**, 2612 (2013).
2. Aran, D., Sirota, M. & Butte, A. J. Systematic pan-cancer analysis of tumour purity. *Nat. Commun.* **6**, 8971 (2015).
3. Saltz, J. *et al.* Spatial Organization and Molecular Correlation of Tumor-Infiltrating Lymphocytes Using Deep Learning on Pathology Images. *Cell Rep.* **23**, 181-193.e7 (2018).
4. Wu, F. *et al.* Single-cell profiling of tumor heterogeneity and the microenvironment in advanced non-small cell lung cancer. *Nat. Commun.* **12**, 2540 (2021).
5. Tirosh, I. *et al.* Dissecting the multicellular ecosystem of metastatic melanoma by single-cell RNA-seq. *Science* **352**, 189–196 (2016).
6. Wu, S. Z. *et al.* A single-cell and spatially resolved atlas of human breast cancers. *Nat. Genet.* **53**, 1334–1347 (2021).
7. Bassez, A. *et al.* A single-cell map of intratumoral changes during anti-PD1 treatment of patients with breast cancer. *Nat. Med.* **27**, 820–832 (2021).
8. Pal, B. *et al.* A single-cell RNA expression atlas of normal, preneoplastic and tumorigenic states in the human breast. *EMBO J.* **40**, (2021).
9. Quinn, T. P., Erb, I., Richardson, M. F. & Crowley, T. M. Understanding sequencing data as compositions: an outlook and review. *Bioinformatics* **34**, 2870–2878 (2018).
10. Lim, E. *et al.* Aberrant luminal progenitors as the candidate target population for basal tumor development in BRCA1 mutation carriers. *Nat. Med.* **15**, 907–913 (2009).
11. Fu, N. Y., Nolan, E., Lindeman, G. J. & Visvader, J. E. Stem Cells and the Differentiation Hierarchy in Mammary Gland Development. *Physiol. Rev.* **100**, 489–523 (2020).

REVIEWER COMMENTS

Reviewer #1 (Remarks to the Author):

I acknowledge the author's earnest efforts in addressing most of my initial concerns. The benchmark study has shown significant improvements. However, there are still a few points that need to be addressed.

Major Comments:

1) Regarding my initial comment (1b), I specifically requested a benchmark study using pseudo bulk data generated from a different scRNA-seq assay, preferably utilizing a distinct scRNA-seq technology, if available. In response, the authors incorporated two additional scRNA-seq datasets for the benchmark study. However, the pseudo-bulk was simulated using the same scRNA-seq dataset used to produce the reference, which means technical batch effects between scRNA-seq and real bulk data could not be accounted for. Considering that the benchmark pipeline has been established, generating such a benchmark study may not require considerable extra effort.

2) In relation to reviewer #3's comment (1b), while conducting the benchmark study using BayesPrism, it appears the authors utilized the updated Gibbs sampling method, which is the software's default setting. However, the developers recommend using the initial Gibbs sampling when there are minimal batch effects, as is the case when both the pseudo-bulk and scRNA-seq reference were generated from the same scRNA-seq dataset. It would be important to adhere to the developers' guidelines in this case.

3) Supplementary Table 3 seems to contain incorrect information, appears incomplete, and lacks consistency with the statistics presented in Extended Data Figure 8d. It lists Scaden and DWLS instead of BayesPrism. The Pearson R statistics for some major cell types are absent, and the term 'median' precedes all statistics without clear context. For example, Extended Data Figure 8d does not seem to match Supplementary Table 3, particularly for 'monocyte, DWLS.': extended data figure 8d (monocyte, DWLS) does not seem to be 0.95. Supplementary Table 3 says 0.3079 which seems to be closer to what is shown in the figure. This discrepancy warrants double-checking. Also, the correlation for 'B cell naïve' in Supplementary Table 3 does not match the figure (in Supplementary Table 3, B cell naïve has a correlation of 0.47, but the figure shows 0.53). It is important to ensure that all figures are correctly labeled.

Minor Comments:

1) I suggest incorporating the full statistics from Rebuttal Figure 1 into the supplement, given the imperfect purity of these methods. For instance, ABSOLUTE requires manual parameter selection, and its different runs can yield substantially varied results, as suggested by Aran et al. I recommend using the CPE score in the main figure.

2) It might be premature to conclude that the use of marker genes does not significantly improve BayesPrism and MuSiC. Given the large sample size, even a small difference in Pearson R can be statistically significant. Marker genes could have a significant, albeit not substantial effect. Conversely, as the authors noted, hspe's performance worsens with marker use, and DWLS is sensitive to marker choice. These results are noteworthy as they can guide users' choices. I suggest summarizing a per-patient correlation for each method for each granularity using a box-plot and presenting it as a supplementary figure.

3) I suggest that the authors double-check the correlation coefficient of Rebuttal Figure 5 for hspe. The "with marker" correlations for minor and subset are considerably worse than the full gene, yet the scatter plot seems similar.

4) The resolution of Figure 5e is quite low, and I suggest replacing it with a high resolution version to improve readability.

Reviewer #2 (Remarks to the Author):

I have thoroughly read the author's response to the reviewer comments, and I have no further comments.

Reviewer #4 (Remarks to the Author): Expert in breast cancer genomics and scRNA-seq; replaces Reviewer #3

As an independent reviewer of the comments and responses to "Reviewer 3", I find that the authors effectively addressed the reviewer comments. Most importantly, the validation studies with Pal and Bassez (Extended data 3 and 4) highlight the generality of the research findings to additional datasets. Further, the addition of TCGA (see responses to Rev. 1) and benchmarking with tumor purity/ABSOLUTE and TIL/Saltz add additional relevant information about performance. In all, the authors have effectively addressed the reviewer concerns in this revised manuscript.

We thank the reviewers for their insightful comments which have further improved our manuscript. Below we provide our response to each comment, with reviewers' comments in bold text.

REVIEWER #1

I acknowledge the author's earnest efforts in addressing most of my initial concerns. The benchmark study has shown significant improvements. However, there are still a few points that need to be addressed.

Major Comments:

1) Regarding my initial comment (1b), I specifically requested a benchmark study using pseudo bulk data generated from a different scRNA-seq assay, preferably utilizing a distinct scRNA-seq technology, if available. In response, the authors incorporated two additional scRNA-seq datasets for the benchmark study. However, the pseudo-bulk was simulated using the same scRNA-seq dataset used to produce the reference, which means technical batch effects between scRNA-seq and real bulk data could not be accounted for. Considering that the benchmark pipeline has been established, generating such a benchmark study may not require considerable extra effort.

Response: We thank the reviewer for highlighting the importance of validating deconvolution performance against technical batch effects. To address this we have conducted an additional benchmarking experiment in which pseudo-bulks were generated using the Bassez et al and Pal et al data, and single-cell reference profiles were generated using the Wu et al data. Of note, the deep learning method Scaden requires pseudo-bulks as training data, which was also generated using scRNA-Seq data from Wu et al.

This analysis required cell-type annotations to be consistent across the three scRNA-Seq datasets. We achieved this by grouping several cell types together for Bassez et al and Pal et al in accordance with the nine major cell types annotated in Wu et al. More specifically, in Bassez et al, we collapsed Tumour-associated macrophages (TAMs) and Dendritic cells (DCs) into Myeloid and dropped Pericytes. In Pal et al, Mast cells and Plasmacytoid dendritic cells (pDC) into Myeloid. Pseudobulk mixtures simulation, deconvolution, and subsequent performance evaluation were conducted using these collapsed cell types.

Results from both Bassez et al and Pal et al showed the same patterns as previous analysis using Wu et al, with BayesPrism, Scaden and MuSiC achieving the lowest Bray-Curtis dissimilarity values across all tumour purity levels.

Changes to manuscripts:

- We have generated a new Supplementary Fig. 10 to show these results. We have altered the numbers of previous Supplementary Figures and references to these figures

have been updated throughout the manuscript.

- We have added the following text in the **Results** section to describe these findings (p.9): *“Additionally, we have tested the robustness of deconvolution performance when technical batch effect is present by using single-cell reference profiles from Wu et al for deconvolution of simulated mixtures generated from Bassez et al. (Supplementary Fig. 10a and b) and Pal et al. (Supplementary Fig. 10c and d). To ensure cell-type labels are consistent across all three datasets, we collapsed several cell types for Bassez et al and Pal et al, e.g. dendritic cell, macrophage and myeloid are grouped into myeloid (see Methods/Datasets and pre-processing). Except for a reduction in performance of hspe in Pal et, results were consistent with patterns observed in Wu at al mixtures (Fig. 2a and b, Supplementary Fig. 4).”*
- We have added the following text to the relevant section in **Discussion** (p.16): *“Similar patterns were also observed when we used simulated mixtures generated from two additional scRNA-Seq datasets, Bassez et al and Pal et al.”*
- In the **Methods** we have updated the section **Simulation of test artificial bulk RNA-Seq mixtures** to describe the processes of intersecting genes and grouping cell-type labels in Bassez et al and Pal et al to ensure consistency with Wu et al.
p.26: *“For the technical batch effect validation experiment where single-cell reference and train simulated mixtures come from Wu et al, we grouped original cell-type labels in Bassez et al and Pal et al consistently with cell-type labels from Wu et al. For Pal et al, we grouped Tumour-associated macrophages (TAMs) and Dendritic cells (DCs) into Myeloid. We also dropped Pericytes, as they are not annotated in Wu et al. This resulted in 8 major cell types: Cancer epithelial, Normal epithelial. T cell, B cell, Myeloid, Endothelial, CAFs, and Plasmablasts. For Bassez et al, we grouped Mast cells and Plasmacytoid dendritic cells (pDC) into Myeloid, resulting in 6 major cell types: Cancer epithelial, T cell, B cell, Myeloid, Endothelial, and CAFs. After this cell annotation grouping step, 57,000 mixtures from Bassez et al, and 38,000 mixtures from Pal et al were generated using the same process described above. For each set of simulated mixtures, we only retained the intersecting genes between Bassez et al and Wu et al, and Pal et al and Wu et al, respectively (Supplementary Table 7).”*
- We have created Supplementary Table 7 to describe intersecting genes between Bassez et al and Wu et al, and between Pal et al and Wu et al.

2) In relation to reviewer #3's comment (1b), while conducting the benchmark study using BayesPrism, it appears the authors utilized the updated Gibbs sampling method, which is the software's default setting. However, the developers recommend using the initial Gibbs sampling when there are minimal batch effects, as is the case when both the pseudo-bulk and scRNA-seq reference were generated from the same scRNA-seq dataset. It would be important to adhere to the developers' guidelines in this case.

We thank the reviewer for highlighting the difference between initial versus final Gibbs sampling results from BayesPrism. We followed up on this suggestion and investigated BayesPrism's deconvolution results before and after Gibbs sampling using pseudobulks generated from both the same single-cell dataset (Wu et al) and different single-cell datasets (Bassez et al and Pal et al). We found that Gibbs sampling significantly improved BayesPrism's performance on cancer and normal epithelial cells, as evidenced by lower cell-type-specific RMSE between heatmaps on the left column compared to heatmaps on the right column in Rebuttal Fig. 1.

Rebuttal Fig. 1. Performance of BayesPrism before and after Gibbs sampling on artificial bulk mixtures generated using scRNA-seq data from Wu et al, Bassez et al and Pal et al. RMSE between predicted and actual cell compositions of artificial bulk mixtures using scRNA-Seq from Wu et al (a, b), Bassez et al (c, d) and Pal et al (e, f), aggregated by cell type. Single-cell

reference profiles and train simulated mixtures (only for Scaden) were from Wu et al, with intersecting genes with Bassez et al and Pal et al when appropriate. Left column (a, c, e) depict BayesPrism performance before Gibbs sampling, while right column (b, d, f) depict performance after Gibbs sampling. Seven tumour purity levels are shown (from 5% to 95%, 15% interval). Darker shade of red represents higher RMSE values (worse performance), with numeric RMSE values shown. CAFs: cancer-associated fibroblast, RMSE: Root Mean Square Error, scRNA-Seq: Single-cell RNA Sequencing.

Interestingly, RMSE values of cancer epithelial cells at 5% tumour purity were smaller before Gibbs sampling compared to after across all three datasets. For pseudobulk from Wu et al and Bassez et al, RMSE values of T-cells and Myeloid were slightly smaller before Gibbs sampling for pseudobulks with tumour purity $\leq 50\%$. These patterns seem to be consistent with BayesPrism's authors guideline that initial cell-type fractions could be more appropriate for mixtures with low tumour fraction ($< 50\%$).

The ability to adjust cell-type fractions using within-sample tumour fraction and across-samples non-tumour fraction is a technical aspect unique only to BayesPrism, which could have contributed to its superior performance over other methods. As the overall performance of BayesPrism was better after Gibbs sampling, we chose to report updated cell-type fractions in the manuscript and have reported the initial cell-type fractions in **Methods** where we describe BayesPrism.

Changes to manuscripts:

- The following text has been added to the relevant section in **Methods**
p.30: “Lastly, BayesPrism produces two sets of predicted cell-type fractions, one before and one after Gibbs sampling step, in which the reference matrix is updated using within-sample tumour expression and across-samples non-tumour expression. We used post-Gibbs-sampling results for BayesPrism in this study and have provided a cross-dataset performance comparison of pre- versus post-Gibbs-sampling in Supplementary Fig. 17.”
- We have included Rebuttal Fig. 1 in the manuscript as Supplementary Fig. 17.

3) Supplementary Table 3 seems to contain incorrect information, appears incomplete, and lacks consistency with the statistics presented in Extended Data Figure 8d. It lists Scaden and DWLS instead of BayesPrism. The Pearson R statistics for some major cell types are absent, and the term 'median' precedes all statistics without clear context. For example, Extended Data Figure 8d does not seem to match Supplementary Table 3, particularly for 'monocyte, DWLS.': extended data figure 8d (monocyte, DWLS) does not seem to be 0.95. Supplementary Table 3 says 0.3079 which seems to be closer to what is shown in the figure. This discrepancy warrants double-checking. Also, the correlation for 'B cell naïve' in Supplementary Table 3 does not match the figure (in Supplementary Table 3, B cell naïve has a correlation of 0.47, but the figure shows 0.53). It is important to ensure that all figures are correctly labeled.

We thank the review for pointing out this critical error and have corrected the data in Supplementary Table 3.

Minor Comments:

1) I suggest incorporating the full statistics from Rebuttal Figure 1 into the supplement, given the imperfect purity of these methods. For instance, ABSOLUTE requires manual parameter selection, and its different runs can yield substantially varied results, as suggested by Aran et al. I recommend using the CPE score in the main figure.

We thank the review for this suggestion and have revised the manuscript accordingly.

Changes to manuscript:

- We have replaced ABSOLUTE results with CPE results in Fig. 1c-d, and incorporated individual results (for ABSOLUTE, ESTIMATE, LUMP and Pathology) within Supplementary Fig 5. CPE results showed the same pattern as ABSOLUTE results, with BayesPrism, MuSiC and Scaden being the top-performing methods at predicting tumour purity.
- We have updated the **Methods** section to reflect this change
p.29: *“We downloaded the Consensus Tumour Estimates (CPE) of 1,023 breast cancer patients in TCGA from Aran et al⁴¹, which was unified across tumour purity estimates by ABSOLUTE⁴⁰, ESTIMATE and LUMP, as well as pathologist-annotated estimates (Pathology) (Supplementary Fig. 16). Out of these 1,023 patients, we were able to match 968 patients back to the downloaded transcriptomics data using patient barcodes. We validated deconvolution performance on purity by comparing predicted cancer epithelial proportions of populations against CPE tumour purity estimations from the filtered list of 968 patients (Supplementary Table 5).”*
- We have also revised the relevant text in **Results** and **Discussion** to reflect the new findings from CPE:
In **Results** (p.9): *“We compared predicted cancer cell proportions against Consensus Purity Estimates (CPE) produced in Aran et al⁴¹...”*
In **Discussion** (p.17): *“Our validation analysis on 1,038 breast cancer samples from TCGA revealed that the three methods showing highest level of resilience against tumour purity (BayesPrism, Scaden and MuSiC) were also the top-performing methods in predicting tumour purity, compared against Aran et al⁴¹ estimates....”*

2) It might be premature to conclude that the use of marker genes does not significantly improve BayesPrism and MuSiC. Given the large sample size, even a small difference in Pearson R can be statistically significant. Marker genes could have a significant, albeit not substantial effect. Conversely, as the authors noted, hspc's performance worsens with marker use, and DWLS is sensitive to marker choice. These results are noteworthy

as they can guide users' choices. I suggest summarizing a per-patient correlation for each method for each granularity using a box-plot and presenting it as a supplementary figure.

Response: We thank the reviewer for this insightful observation. We have followed up on the reviewer's suggestion and reworked the visualizations summarizing with-marker-genes versus no-marker-genes comparisons for each method.

To measure overall performance, we computed mixture-to-mixture Bray-Curtis dissimilarity and displayed the distributions of each method at each lineage level in boxplots (Supplementary Fig. 18a). Note here we only compared with-marker-genes versus no-marker-genes performance only within each lineage level and not across lineages, hence the use of Aitchison distance was not needed. In addition, we have also computed the mixture-to-mixture Root-mean-square-error and summarized this information in Supplementary Fig. 18b, which shows the same patterns as Supplementary Fig. 18a.

Changes to manuscript:

- We have updated the relevant text in the **Methods** section to highlight that marker genes could potentially have an impact on deconvolution performance.
p.30: *"The methods BayesPrism, Bisque, MuSiC and hspe could incorporate cell-type-specific marker genes during execution. By contrast, DWLS could either use Seurat or MAST to build its internal signature matrix. In this study, we chose to use default parameters for all methods and hence reported results produced by the no-marker-genes version of BayesPrism, Bisque, MuSiC and hspe, and the Seurat version of DWLS. We have, however, provided a performance comparison across all three immune lineage levels in Supplementary Fig. 18, which shows slight differences for BayesPrism, Bisque, MuSiC and better performance for hspe when marker genes are not used (Supplementary Fig. 18a-b), as well as slightly better performance for Seurat-DWLS (Supplementary Fig. 18c-d). When performance optimisation is of concern, we do recommend considering different parameter settings for each method."*

3) I suggest that the authors double-check the correlation coefficient of Rebuttal Figure 5 for hspe. The "with marker" correlations for minor and subset are considerably worse than the full gene, yet the scatter plot seems similar.

We thank the review for pointing out this error. As mentioned in our response to the previous comment, we have replaced Pearson's r with Bray-Curtis dissimilarity and RMSE to measure overall performance of with-marker-genes vs no-marker-genes (Supplementary Fig. 18).

4) The resolution of Figure 5e is quite low, and I suggest replacing it with a high resolution version to improve readability.

We have re-saved all images in vector format (.svg) and re-edited all figures to include only .svg files, including Fig. 5e.

REVIEWER #2

I have thoroughly read the author's response to the reviewer comments, and I have no further comments.

We thank the reviewer for their previous helpful suggestions.

REVIEWER #4: Expert in breast cancer genomics and scRNA-seq; replaces Reviewer #3

As an independent reviewer of the comments and responses to "Reviewer 3", I find that the authors effectively addressed the reviewer comments. Most importantly, the validation studies with Pal and Bassez (Extended data 3 and 4) highlight the generality of the research findings to additional datasets. Further, the addition of TCGA (see responses to Rev. 1) and benchmarking with tumor purity/ABSOLUTE and TIL/Saltz add additional relevant information about performance. In all, the authors have effectively addressed the reviewer concerns in this revised manuscript.

We thank Reviewer #4 for taking over from Reviewer #3.

REVIEWERS' COMMENTS

Reviewer #1 (Remarks to the Author):

The authors have addressed all my comments. I appreciate their efforts. In addition, I suggest that the authors consider replacing the supplementary figures with higher resolution versions during the typesetting process.